# IFNγ causes mitochondrial dysfunction and oxidative stress in myositis

Catalina Abad [1], Iago Pinal-Fernandez [2,3], Clement Guillou[4],
Gwladys Bourdenet [1], Laurent Drouot[1], Pascal Cosette[4,5], Margherita Giannini[6,7],
Lea Debrut [6], Laetitia Jean[1], Sophie Bernard [8], Damien Genty[9],
Rachid Zoubairi[1], Isabelle Remy-Jouet[10], Bernard Geny[6,7], Christian Boitard[11],
Andrew Mammen [2,3,12], Alain Meyer[6,7] & Olivier Boyer [1,13] ✉

Idiopathic inflammatory myopathies (IIMs) are severe autoimmune diseases with poorly understood pathogenesis and unmet medical needs. Here, we examine the role of interferon γ (IFNγ) using NOD female mice deficient in the inducible T cell co-stimulator (*Icos*), which have previously been shown to develop spontaneous IFNγ-driven myositis mimicking human disease. Using muscle proteomic and spatial transcriptomic analyses we reveal profound myofiber metabolic dysregulation in these mice. In addition, we report muscle mitochondrial abnormalities and oxidative stress in diseased mice. Supporting a pathogenic role for oxidative stress, treatment with a reactive oxygen species (ROS) buffer compound alleviated myositis, preserved muscle mitochondrial ultrastructure and respiration, and reduced inflammation. Mitochondrial anomalies and oxidative stress were diminished following anti-IFNγ treatment. Further transcriptomic analysis in IIMs patients and human myoblast in vitro studies supported the link between IFNγ and mitochondrial dysfunction observed in mice. These results suggest that mitochondrial dysfunction, ROS and inflammation are interconnected in a self-maintenance loop, opening perspectives for mitochondria therapy and/or ROS targeting drugs in myositis.

Idiopathic inflammatory myopathies (IIMs) are a group of severe acquired autoimmune skeletal muscle diseases characterized by progressive muscle damage and weakness[1]. Multiple lines of evidence suggest IIMs are immune-mediated diseases. Depending on the IIM type, these may include the presence of specific autoantibodies in serum, immune cell infiltrates in muscle biopsies, activation of the complement cascade or upregulation of major histocompatibility complex molecules by muscle fibers[2]. In addition, upregulation of interferon (IFN)-induced genes has been reported in myositis muscle biopsies, with type I or II IFN pathways being preferentially activated in

[1]Univ Rouen Normandie, Inserm, UMR1234, FOCIS Center of Excellence PAn'THER, F-76000 Rouen, France. [2]Muscle Disease Unit, National Institute of Arthritis and Musculoskeletal and Skin Diseases, National Institutes of Health, Bethesda, MD, USA. [3]Department of Neurology, Johns Hopkins University School of Medicine, Baltimore, MD, USA. [4]Univ Rouen Normandie, Inserm US 51, CNRS UAR 2026, HeRacLeS PISSARO, F-76000 Rouen, France. [5]Univ Rouen Normandie, INSA Rouen Normandie, CNRS, Normandie Univ, PBS UMR 6270, F-76000 Rouen, France. [6]Translational Medicine Federation of Strasbourg, Team 3072, Faculty of Medicine, University of Strasbourg, Strasbourg, France. [7]Unité exploration fonctionnelle musculaire-service de physiologie, Centre National de Référence des Maladies Auto-Immunes Systémiques Rares de l'Est et du Sud-Ouest -Service de rhumatologie, Hôpitaux Universitaires de Strasbourg, Strasbourg, France. [8]Univ Rouen Normandie, Inserm US51, CNRS UAR2026, HeRacLeS PRIMACEN, F-76000 Rouen, France. [9]CHU Rouen, Department of Pathology, F-76000 Rouen, France. [10]Univ Rouen Normandie, Inserm, UMR1096, BOSS facility, F-76000 Rouen, France. [11]Cochin Institute, Paris Descartes University, Sorbonne Paris Cité, Paris, France. [12]Department of Medicine, Division of Rheumatology, Johns Hopkins University School of Medicine, Baltimore, MD, USA. [13]CHU Rouen, Department of Immunology and Biotherapy, F-76000 Rouen, France. ✉e-mail: Olivier.boyer@chu-rouen.fr

myositis subtypes[3,4]. Glucocorticoids are the anchor drugs of initial treatment together with immunosuppressive agents such as methotrexate or azathioprine[5,6]. Nevertheless, the prolonged use of high-dose corticosteroids is associated with severe adverse events, including acquired myopathy. Other immunosuppressants (e.g., mycophenolate mofetil, cyclophosphamide) or biological drugs targeting immune responses (e.g., rituximab) may be used in refractory or severe patients[6]. Moreover, administration of intravenous immune globulin (IVIG), a therapy approved for certain autoimmune conditions, has shown benefits in dermatomyositis (DM) clinical trials[7,8]. Nevertheless, the therapeutic efficacy of these drugs may reveal insufficient and IIMs remain conditions with unmet medical needs. Non-immune mechanisms may contribute to the chronicity of IIMs, although these have been far less investigated[9]. For example, endoplasmic reticulum stress has been implicated in the perpetuation and aggravation of the disease[10]. Mitochondrial abnormalities have been observed in myositis but the link with the muscle inflammatory process has been partially investigated[11–15]. Elucidating the pathophysiological pathways in IIMs and targeting these mechanisms may help ameliorating the management of myositis.

The scarcity of myositis animal models has been a challenge in understanding the pathophysiology of IIMs and for therapeutic drug discovery. Experimental autoimmune myositis (EAM) models based on immunization with muscle homogenates, myosin, or C-protein may be useful to unravel certain aspects of the disease, but their specific immunogen-driven nature may not reflect the complexity of IIMs mechanisms[16]. We previously reported the occurrence of severe spontaneous muscle autoimmune disease in Non-Obese Diabetic (NOD) mice with a constitutive deletion of the costimulatory molecule ICOS or its ligand (*Icos*[-/-] NOD or *Icosl*[-/-] NOD mice)[17–19]. Whereas NOD mice constitute a classical model of spontaneous type 1 diabetes, *Icos*[-/-] NOD and *Icosl*[-/-] NOD mice exhibit progressive and chronic muscle inflammation with Th1-infiltrating cells, while being protected from diabetes development. Adoptive transfer experiments confirmed that *Icosl*[-/-] NOD mice pathology is CD4[+] T cell-dependent[17].

Here, we examine the mechanisms involved in *Icos*[-/-] NOD myositis and report the existence of severe mitochondrial defects and the beneficial effects of reactive oxygen species (ROS)-buffer administration. By means of transcriptome data analysis of muscle biopsies from patients with DM, and in vitro studies of human myoblasts exposed to IFNγ, we confirmed the correlation between this cytokine and mitochondrial anomalies in human myositis.

## Results

### *Icos*[-/-] NOD mice exhibit inflammatory myopathy with profound muscle metabolic imbalance

Myopathy in *Icos*[-/-] NOD mice is age-dependent, with progressive muscle immune cell infiltration observed from 25 weeks of age (Fig. 1a and Supplementary Fig. 1). Consistent with the type I and/or II IFN signatures in human IIMs[4] and the Th1 cell involvement in *Icosl*[-/-] NOD myositis[17], *Ifnb* and *Ifng* mRNA gene expressions were significantly elevated in *Icos*[-/-] NOD mouse muscle, with higher levels of *Ifng* (Fig. 1b). Expression of genes coding for the IFN-driven chemokines CXCL9 (MIG) and CXCL10 (IP-10) as well as that of CCL2 (MCP-1) (*Ccl2*, *Ccl9*, and *Cxcl10* genes) were also upregulated (Fig. 1b).

To further understand the mechanisms involved in *Icos*[-/-] NOD mice myopathy, we performed muscle holoproteome analysis at 8 (absence of clinical disease), 25 (onset), and 35 weeks of age (established myopathy) (Fig. 1c–g). Hierarchical clustering revealed changes in relative protein abundances across the *Icos*[+/+] NOD and the *Icos*[-/-] NOD genotypes, with mice of the same age and genotype in close proximity in terms of proteomic signature profiles (Supplementary Fig. 2a). Further analysis revealed abnormal levels of multiple proteins in *Icos*[-/-] NOD vs. *Icos*[+/+] NOD mice at all ages (148, 254, and 254 proteins, at 8, 25, and 35 weeks of age, respectively), most of them being

downregulated (Fig. 1c). Indeed, the levels of more than 200 proteins were significantly reduced in mice exhibiting myopathy (241 at 25 weeks of age and 219 at 35 weeks of age) (Fig. 1c). We performed a Protein–Protein Interaction (PPI) analysis of deregulated proteins using the STRING network, identifying functional enrichments based on Gene Ontology-Biological Process (GO-BP) and Reactome classification systems (Fig. 1d–f). Non-surprisingly, some upregulated proteins in the muscles from 25- and 35-week-old *Icos*[-/-] NOD mice were immune-related proteins (Fig. 1e). Strikingly, the majority of the downregulated proteins (over 60%) were proteins pertaining to metabolic pathways including fatty acid, nucleotide and carbohydrate metabolism, tricarboxylic acid (TCA) cycle, oxidative phosphorylation (OXPHOS) and protein synthesis/folding (Fig. 1d–f). 'Response to oxidative stress' proteins were also identified (Fig. 1d). Remarkably, certain metabolic alterations were already present in mice at an early age, *i.e.* in asymptomatic mice. STRING proteome data analysis classifying dysregulated proteins based on cellular components to which they belong clearly revealed that a greater number of the altered proteins in *Icos*[-/-] NOD mouse muscles was ascribed to mitochondria (Fig. 2a). This suggests metabolic deregulation and mitochondrial abnormalities since an early phase of the disease, preceding conspicuous immune cell muscle infiltration.

Further in-depth Ingenuity Pathway Analysis (IPA) situated 'oxidative phosphorylation' on top of the list of downregulated pathways in *Icos*[-/-] NOD mice and included other mitochondrial metabolic pathways such as the 'TCA cycle' and 'fatty acid beta-oxidation' (Fig. 1g and Supplementary Fig. 2b). Within the 109 proteins of the 'oxidative phosphorylation' pathway, 7 (6.42%), 21 (19.26%), and 21 (19.26%) were dysregulated in 8-, 25- and 35-week-old *Icos*[-/-] NOD mice, respectively (Supplementary Fig. 2c). The levels of multiple proteins belonging to the respiratory chain complexes I (NADH dehydrogenase), II (succinate dehydrogenase, SDH), III (cytochrome bc1 complex), IV (cytochrome C oxidase, COX), and V (ATP synthase) were significantly reduced at all ages in the muscle of *Icos*[-/-] NOD mice compared to *Icos*[+/+] NOD mice (Fig. 2b, c). This result was evocative of mitochondrial alterations. Notably, a certain number of non-immune proteins was upregulated in the muscle of 8-week-old mice including PRDX5 (peroxiredoxin 5), an antioxidant enzyme whose levels are known to raise upon oxidative stress (Fig. 2b)[20]. These increases might reflect an attempt to dampen incipient mitochondrial defects and oxidative stress.

### *Icos*[-/-] NOD mice exhibit muscle mitochondrial morphological and functional abnormalities

Assessing the activity of respiratory enzymes (i.e. COX, NADH-TR, and SDH) through muscle histoenzymology stainings is a straightforward approach for investigating mitochondrial dysfunction and is used for IIM diagnosis. Myofibers with high oxidative capacity (slow-twitch oxidative or Type 1) are rich in mitochondria and stain darker for these enzymes than glycolytic (Type 2) fibers. In contrast to the typical normal mosaic pattern of dark and light fibers in *Icos*[+/+] NOD mice, profound alterations in COX, NADH-TR, and SDH staining patterns were found in muscle sections from *Icos*[-/-] NOD mice with myopathy, particularly in mice with established disease (Fig. 3a). Loss of enzymatic activity, with pale fibers and white spots within otherwise homogeneously dark fibers, implied mitochondrial deficiencies (Fig. 3a). Indeed, there was a significant loss of myofibers with high COX activity in mice with established disease (Fig. 3a). Of note, the patterns of COX, SDH, and NADH activities were identical among each other and we did not observe any SDH-positive fibers that were COX-negative in *Icos*[-/-] NOD mice (Fig. 3a), a trait characteristic of human primary mitochondrial myopathies due to mtDNA mutations[21].

We next investigated mitochondrial respiration in skinned quadriceps muscle fibers ex vivo. The respiratory control ratio (RCR),

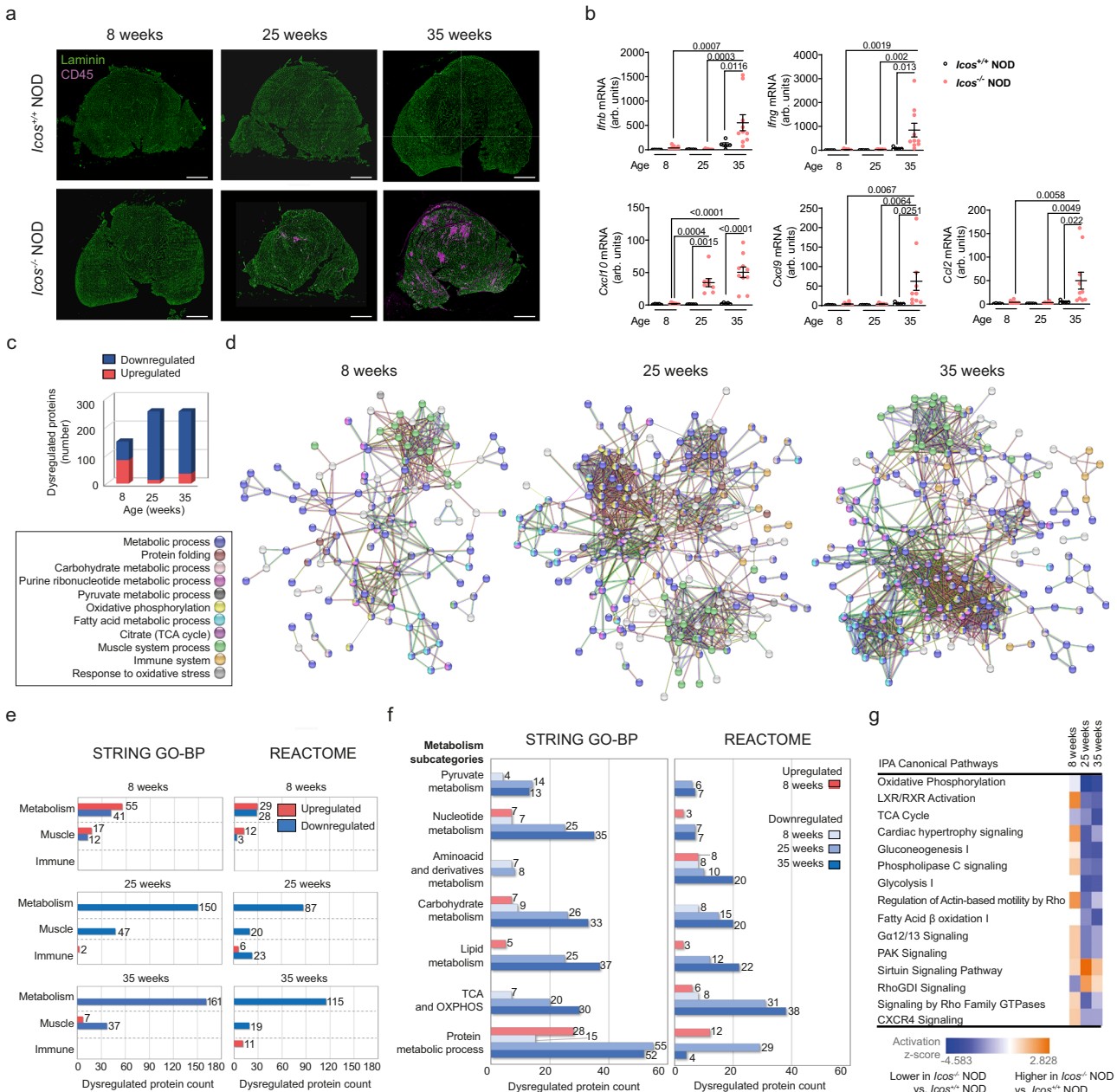

**Fig. 1 | *Icos⁻/⁻* NOD mice developing myositis exhibit profound muscle metabolic disturbances. a** Immunofluorescence staining of immune cells (CD45) and myofibers (laminin) in the muscles of *Icos⁺/⁺* NOD and *Icos⁻/⁻* NOD mice of 8, 25, and 35 weeks of age (predisease, onset and established disease in the *Icos⁻/⁻* NOD mice, respectively). A representative image out of *n* = 10 independent mice/genotype is shown. Scale bar, 1 mm. **b** Cytokine (*Ifng* and *Ifnb*) and chemokine (*Ccl2*, *Cxcl9*, *Cxcl10*) mRNA gene expression levels in the muscles of *Icos⁺/⁺* NOD and *Icos⁻/⁻* NOD mice at different ages; for *Icos⁺/⁺* NOD at all ages *n* = 5 independent mice/group; for *Icos⁻/⁻* NOD mice, *n* = 8 (8 and 25 weeks of age) and *n* = 10 (35 weeks of age) independent mice/group (arb. units: arbitrary units). Numbers denote *p* values. Mean values ± s.e.m are shown. **a**, **b** were repeated independently twice with one representative experiment being shown. **c–g** Proteome analysis of *Icos⁻/⁻* NOD vs. *Icos⁺/⁺* NOD mice muscles at 8, 25, and 35 weeks of age (*n* = 5 independent mice/genotype and age). **c** Number of dysregulated proteins in *Icos⁻/⁻* NOD vs. *Icos⁺/⁺* NOD muscles. **d** Protein-protein interaction (PPI) map of dysregulated proteins in *Icos⁻/⁻*

NOD vs. *Icos⁺/⁺* NOD mice muscles, highlighting major altered pathways (input data corresponds to the mean of *n* = 5 independent mice) (STRING Gene ontology-Biological Process (GO-BP) analysis for all except for 'Immune System', which protein interactions were only identified by Reactome analysis). **e** Graphs showing the percentage of enriched STRING GO-BP and Reactome terms of the total terms for metabolic, muscle, and immune processes. **f** Proteins that are involved in 'Metabolism' (according to STRING GO-BP) were subdivided into the indicated categories based on STRING GO-BP or Reactome database analysis. **g** Top 15 'Canonical pathways' identified using Ingenuity Pathway Analysis (IPA) of dysregulated proteins of *Icos⁻/⁻* NOD vs. *Icos⁺/⁺* NOD mice. For **b**, statistical analyses were performed using the Two-way ANOVA test with Sidak's post-hoc multiple comparisons. For the identification of significantly dysregulated proteins, statistical analysis was performed using the inbuilt Progenesis statistical box called 'one-way ANOVA'. For the IPA analysis, Fisher's Exact test was used. Source data are provided as a Source Data file.

representing the ratio of oxygen uptake used for ATP formation (OXPHOS by CI) with respect to the oxygen uptake required to compensate for proton leakage, was significantly reduced in the *Icos⁻/⁻* NOD mice with established disease (Fig. 3b). This indicated a diminished

efficiency of mitochondrial OXPHOS and thus impaired mitochondrial respiration in *Icos⁻/⁻* NOD muscle.

Finally, we investigated whether the loss of mitochondrial function was associated with morphological alterations by electron

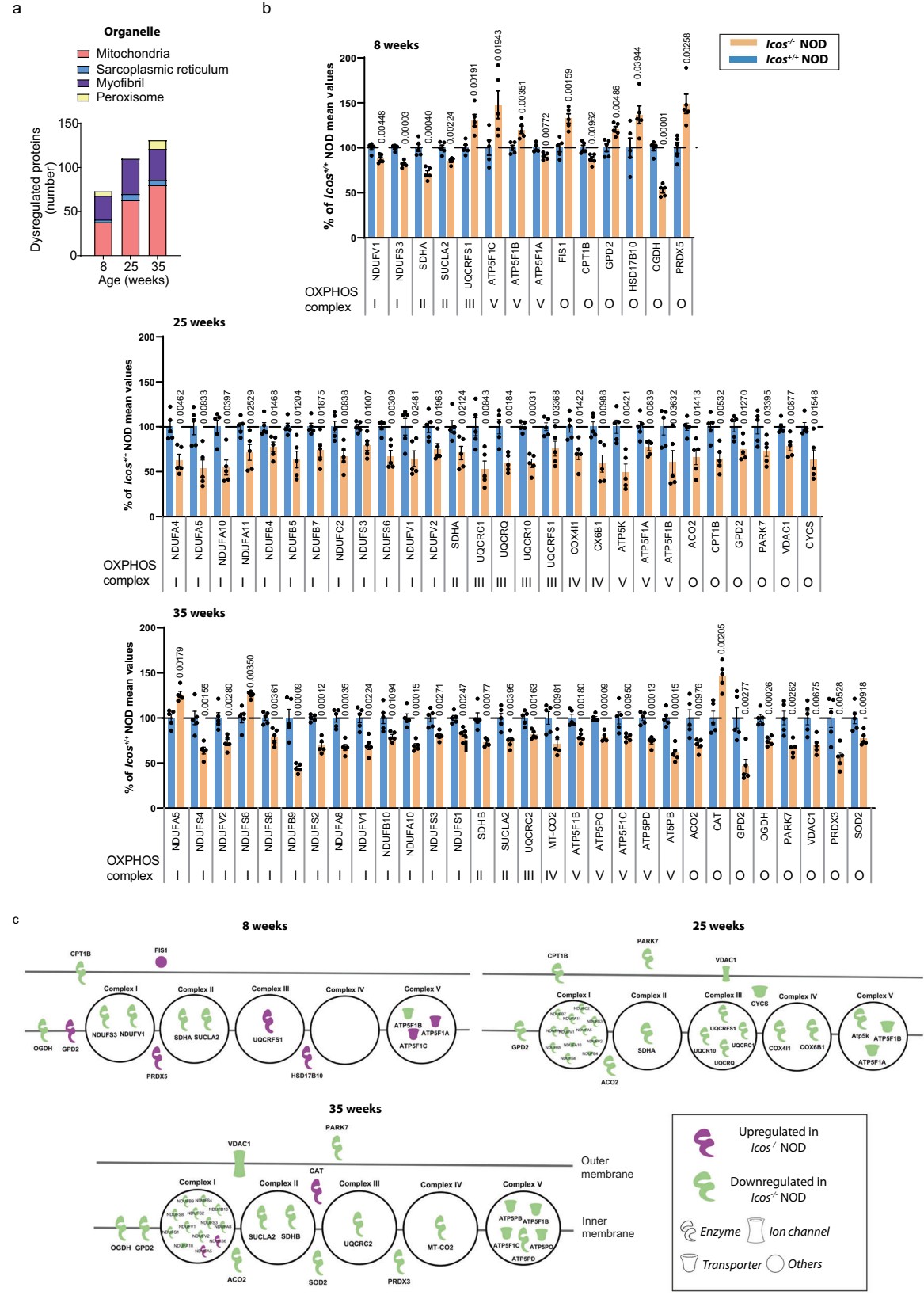

**Fig. 2 | Proteome analysis of muscle from *Icos*[-/-] NOD mice developing myositis reveals the presence of mitochondrial anomalies. a** In-depth STRING proteome analysis of *Icos*[-/-] NOD vs. *Icos*[+/+] NOD mice muscles showing identified dysregulated proteins classified by organelle. **b** IPA proteome analysis showing altered proteins belonging to the mitochondrial respiratory chain in the muscles of *Icos*[-/-] NOD vs. *Icos*[+/+] NOD mice. Histograms represent normalized protein levels with respect to *Icos*[+/+] NOD mice (100%) (*n* = 5 independent mice/group). Mean values ± s.e.m are shown. **c** IPA-generated schematic graphic representing dysregulated protein distribution in the mitochondrial respiratory chain (respiratory chain complexes I to V and O, for other). Statistical analysis was performed using the inbuilt Progenesis statistical box called 'one-way ANOVA'. Numbers on graphs denote *p* values obtained by comparison of *Icos*[-/-] NOD vs. *Icos*[+/+] NOD raw data. Source data are provided as a Source Data file.

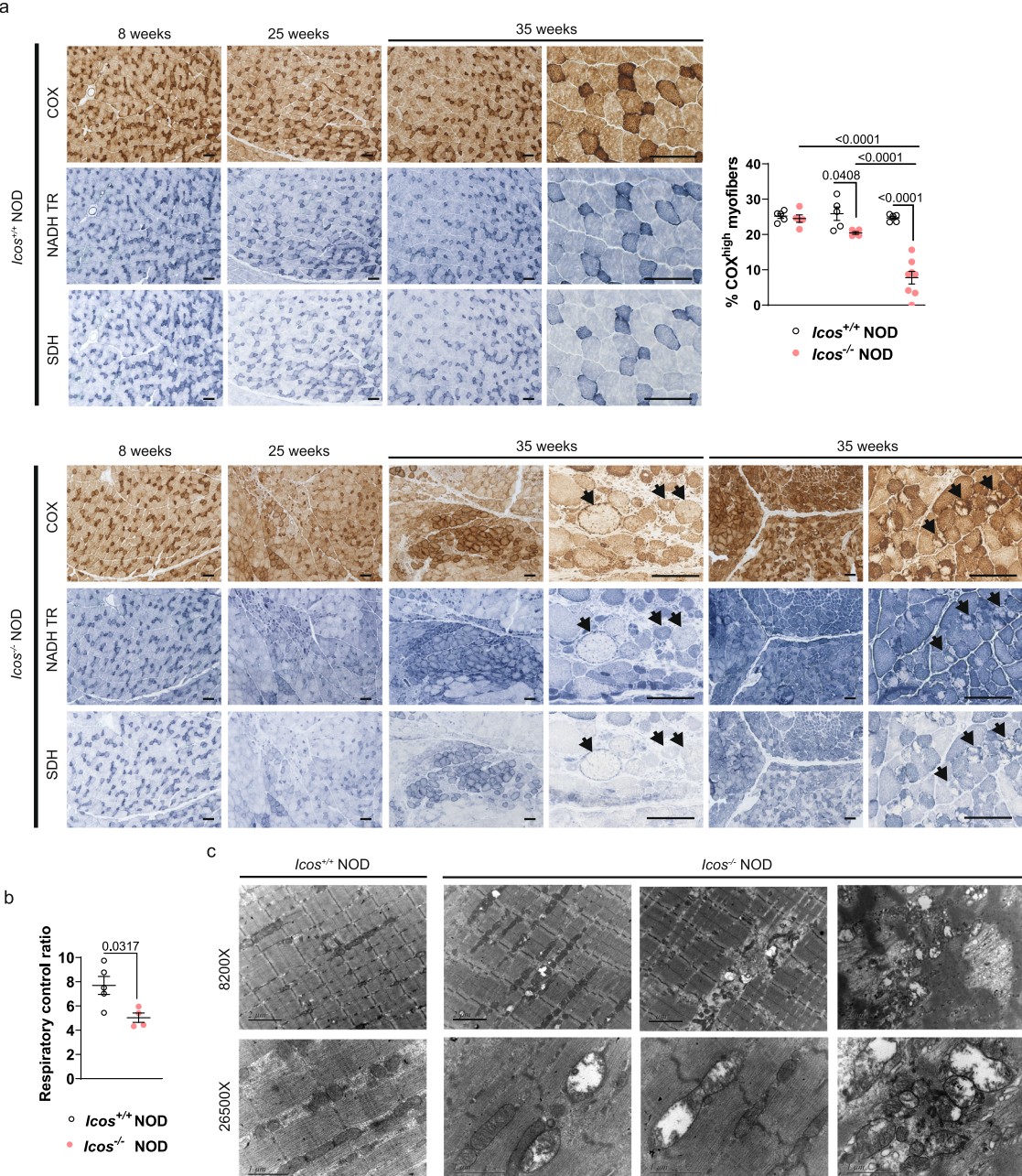

**Fig. 3 | Mitochondrial functional and morphological anomalies are present in the muscles of *Icos*-/- NOD mice. a** Histoenzymology COX, NADH-TR, and SDH stainings of *Icos*-/- NOD and *Icos*+/+ NOD mice muscles at 8, 25, and 35 weeks of age (predisease, onset, and established disease stages) (representative images from *n* = 10 independent mice/genotype and age analyzed). For *Icos*-/- NOD mice at 35 weeks of age, two lower and two higher magnification representative images are shown. Arrows point at pale myofibers or dark fibers exhibiting loss of staining areas. Scale bars, 100 μm. Quantification of COX^high fibers on images of entire quadriceps sections is shown on the right graph (*n* = 5 independent mice/

genotype/age except for *Icos*-/- NOD mice at 35 weeks of age, with *n* = 8 independent mice). Statistical analysis was performed using the Two-way ANOVA test with Sidak's post-hoc multicomparison. **b** Ex vivo analysis of oxygen consumption (respiratory control ratio) in *Icos*-/- NOD (*n* = 4 independent mice) and *Icos*+/+ NOD mice (*n* = 5 independent mice) skinned muscles fibers. Statistical analyses were performed using the Mann–Whitney test (two-tailed). Mean values ± s.e.m are shown. **c** Electron microscopy representative images of muscles from *Icos*-/- NOD and *Icos*+/+ NOD mice. **a**–**c** were repeated independently twice. Numbers on panels denote *p* values. Source data are provided as a Source Data file.

microscopy. Strikingly, we found a high prevalence of mitochondria with profound morphological abnormalities such as swelling and loss of matrix and crests (Fig. 3c).

**Spatial transcriptomics reveals an association between inflammatory infiltrates and reduced mitochondrial metabolism**

To dissect the relationship between mitochondrial impairment and immune cell infiltration, we performed spatial transcriptome analysis to compare the gene signatures of *Icos*+/+ NOD vs. *Icos*-/- NOD myofibers

either in direct contact or not with immune cell infiltrates. Segmentation using specific markers, allowed the collection and analysis of mRNAs from myofibers (desmin+) excluding immune cells (CD45+) of the same area (Fig. 4a). Hence, we analyzed the whole transcriptome of different areas of illumination (AOI): *Icos*-/- NOD myofibers in proximity, but not directly adjacent, to a large cluster of infiltrating immune cells (*Icos*-/- NOD PROX), myofibers directly adjacent to a large infiltrating immune cell cluster (*Icos*-/- NOD ADJ) and myofibers from *Icos*+/+ NOD mice (Fig. 4a). Principal components analysis (PCA) and t-distributed

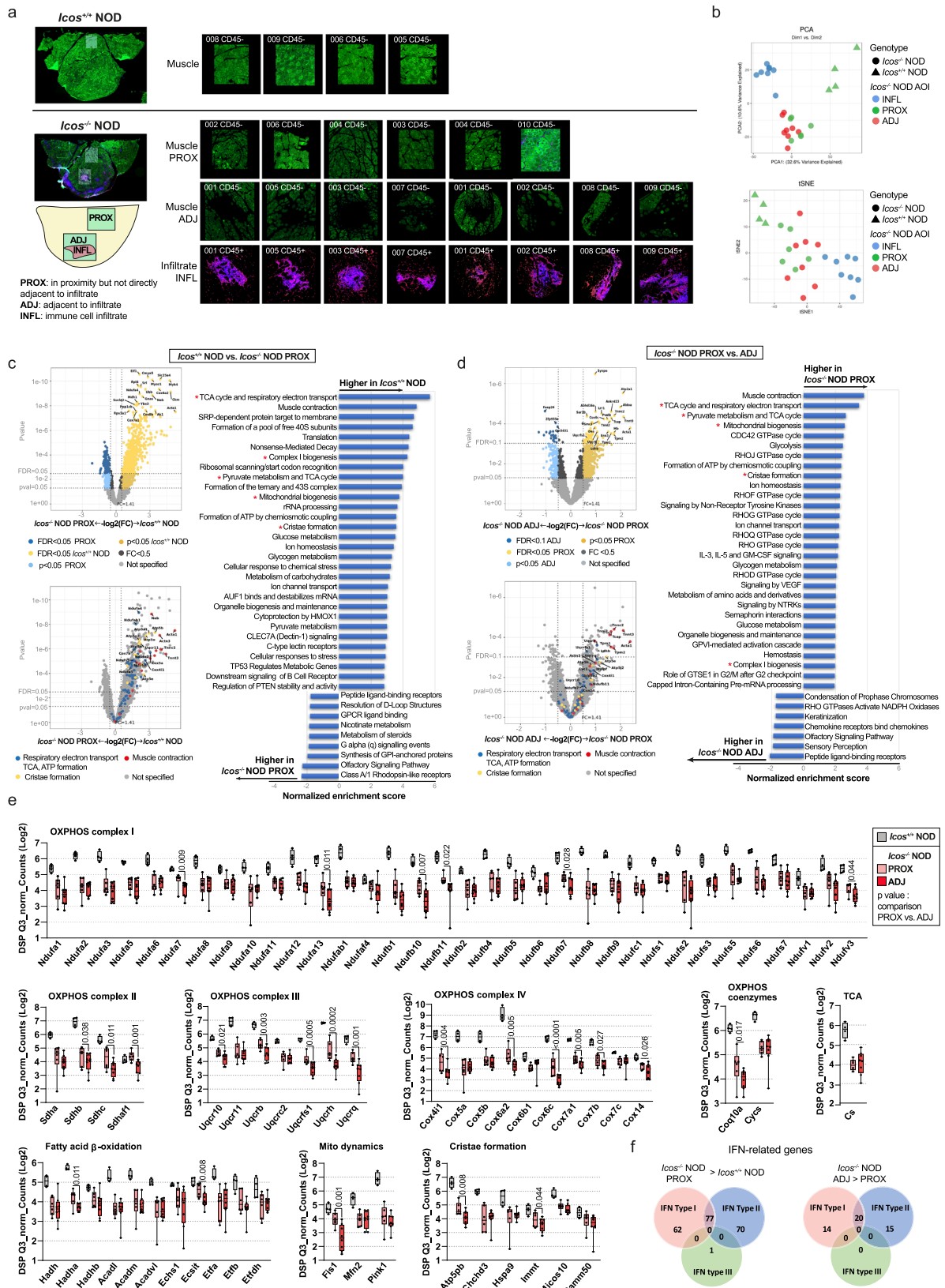

stochastic neighbor embedding (t-SNE) dimensionality reduction analysis of normalized expression showed clustering of the samples corresponding to each group (Fig. 4b). We compared the transcriptomic signatures of *Icos*⁺/⁺ NOD mice vs. *Icos*⁻/⁻ NOD PROX myofibers using LMM statistical analysis (Fig. 4c). In this first comparison,

the pool of genes with the highest statistical differences included muscle (e.g. *Acta1*, *Myh4*, *Myoz1*) and mitochondrial (e.g. *Cox6a2*, *Cox7a*, *Cox8b*, *Ndufa4*, *Slc25a4*) genes, being reduced in *Icos*⁻/⁻ NOD PROX vs. *Icos*⁺/⁺ NOD mice myofibers (Fig. 4c). Reactome pathway analysis confirmed the strong downregulation of 'muscle contraction'

**Fig. 4 | Spatial transcriptome analysis were evocative of mitochondrial impairments in myofibers from *Icos⁻/⁻* NOD mice.** NanoString GeoMx Digital Spatial Profiling was used to perform whole transcriptome analysis. **a** Areas of illumination (AOIs) on the basis of morphology markers (CD45 and desmin). From *n* = 4 *Icos⁺/⁺* NOD mice, *n* = 4 Regions Of Interest (ROIs, 1 ROI/mouse) for a total of *n* = 4 independent desmin⁺CD45⁻ AOIs. From *n* = 6 *Icos⁻/⁻* NOD mice, two types of ROIs were selected according to their distance to an immune cell infiltrate: *n* = 6 ROIs (1 ROI/mouse) in proximity but not adjacent to an immune infiltrate cluster for a total of *n* = 6 independent desmin⁺CD45⁻ AOIs (*Icos⁻/⁻* NOD PROX myofibers), and *n* = 8 ROIs (for 4 mice, 1 ROI/mouse and for 2 mice, 2 ROI/mouse) from *Icos⁻/⁻* NOD mice adjacent to a large immune cell infiltrate cluster for a total of *n* = 8 independent desmin⁺CD45⁻ AOIs (*Icos⁻/⁻* NOD ADJ myofibers) and *n* = 8 independent desmin⁻CD45⁺ AOIs (*Icos⁻/⁻* NOD INF). **b** Principal component analysis (PCA) and t-Distributed Stochastic Neighbor Embedding (t-SNE) based on gene expression data of all AOIs. Each symbol represents one independent AOI. **c** Volcano plots and pathway analysis from statistical comparisons between *Icos⁺/⁺* NOD vs. *Icos⁻/⁻* NOD PROX myofibers or **d**, *Icos⁻/⁻* NOD PROX vs. ADJ myofibers. **e** Histograms depicting normalized RNA counts for genes related to mitochondrial metabolism/structure (*n* = 4 independent AOIs from 4 *Icos⁺/⁺* NOD mice, *n* = 6 independent AOIs '*Icos⁻/⁻* NOD PROX' from 6 *Icos⁻/⁻* NOD mice and *n* = 8 independent AOIs '*Icos⁻/⁻* NOD ADJ' from the same 6 *Icos⁻/⁻* NOD mice). Box plots bounds to 25th to 75th percentiles, with line at the median, and whiskers expand from min to max values. **f** Venn diagrams depicting dysregulated genes reported to be modulated by Type I, II and II IFNs as identified by the Interferome database. For **c** and **d**, red asterisks high-light mitochondria-related pathways. Q3 normalization was used for all data and a Linear Mixed Model (LMM) with Bonferroni-Hochberg (BH) correction was used for statistical analysis. Non adjusted *p* values of comparison between *Icos⁻/⁻* NOD PROX and ADJ are depicted. For pathway analysis, Gene Set Enrichment Analysis (GEA) was performed. Source data are provided as a Source Data file.

genes (normalized enrichment score (NES) 4.84, adjusted *p*-value 0.002) (Fig. 4c and Supplementary Fig. 3). Moreover, mitochondrial pathways were downregulated in *Icos⁻/⁻* NOD PROX myofibers, with metabolism-related pathways being on top of the list ('TCA cycle and respiratory electron transport', NES 5.67, adjusted *p*-value 0.002), but also 'mitochondrial biogenesis' (NES 3.76, adjusted *p*-value 0.002) and 'cristae formation' (NES 3.57, adjusted *p*-value 0.002). These categories also appeared when comparing the gene expression profiles of *Icos⁺/⁺* NOD mice vs. *Icos⁻/⁻* NOD ADJ myofibers (Supplementary Fig. 4). Indeed, the pool of differentially expressed genes between these two groups also included muscle and mitochondrial genes (Supplementary Fig. 4). Interestingly, when comparing the transcriptomic profile of *Icos⁻/⁻* NOD PROX vs. ADJ myofibers, we observed a differential gene expression that also applied to metabolic pathways and mitochondrial stability (Fig. 4d). In addition to OXPHOS genes, histograms depict normalized mRNA counts for all genes from 'TCA and respiratory electron transport', 'fatty acid β-oxidation' and 'cristae formation' pathways, which were downregulated in both *Icos⁻/⁻* NOD PROX and ADJ myofibers compared to *Icos⁺/⁺* NOD myofibers (Fig. 4e). In addition, we observed in *Icos⁻/⁻* NOD myofibers a significant reduction of mRNA counts for genes coding for proteins involved in mitochondrial dynamics regulation through fission/fusion (*Pink1*, *Fis1*, and *Mfn2*), which are mitochondrial quality control mechanisms (Fig. 4e).

In agreement with the observation of mitochondrial ultra-structural anomalies, we found in *Icos⁻/⁻* NOD ADJ and PROX myofibers, a reduction of mRNAs coding for citrate synthase (*Cs*), which is commonly used as a quantitative marker for the presence of intact mito-chondria, as well as of genes involved in cristae formation (*Atp5pb*, *Chchd3*, *Hspa9*, *Immt*, *Micos10*, *Samm50*; Fig. 4e). Accordingly, our bulk muscle tissue holoproteome analysis showed that the proteins levels of citrate synthase (CISY), as well as of Mitochondrial contact site and Cristae Organizing System (MICOS) 19, MICOS26 and Sorting and Assembly Machinery (SAM) 50, three proteins with a crucial role in the maintenance of the structure of mitochondrial cristae and the proper assembly of the mitochondrial respiratory chain complexes[22], were also decreased in *Icos⁻/⁻* NOD diseased mice (Supplementary Fig. 5).

The expression of certain mitochondrial genes in *Icos⁻/⁻* NOD myofibers exhibited a gradual decrease towards the proximity with the immune infiltrate, while others seemed equally reduced in *Icos⁻/⁻* NOD myofibers from either PROX or ADJ zones (Fig. 4e). Indeed, the mRNA levels of 26 out of the 78 mitochondrial genes in Fig. 4e (*Ndufa7*, *Ndufa13*, *Ndufb7*, *Ndufb10*, *Ndufb11*, *Ndufv3*, *Sdhb*, *Sdhc*, *Sdhaf1*, *Uqcr10*, *Uqcrb*, *Uqcrfs1*, *Uqcrh*, *Uqcrq*, *Cox4i1*, *Cox6a2*, *Cox6c*, *Cox7a1*, *Cox7b*, *Cox14*, *Coq10a*, *Hadha*, *Ecsit*, *Fis1*, *Atp5pb*, *Immt*) were lower in *Icos⁻/⁻* NOD ADJ vs. PROX myofibers.

Because of the involvement of type I and type II IFNs in the pathophysiology of IIMs, and the high *Ifnb* and *Ifng* gene expressions in whole muscle from *Icos⁻/⁻* NOD mice, we interrogated the presence of IFN-related genes within the list of genes that are statistically higher in *Icos⁻/⁻* NOD PROX vs. *Icos⁺/⁺* NOD myofibers or in *Icos⁻/⁻* NOD ADJ vs. PROX myofibers using the open access database of types I, II, and III interferon regulated genes Interferome v2.0. This analysis revealed enhanced mRNA expression of multiple genes associated with the interferon response within *Icos⁻/⁻* NOD myofibers (Fig. 4f).

Overall, spatial transcriptome analysis supports the presence of mitochondrial anomalies in *Icos⁻/⁻* NOD mice myofibers, with accent-uation in myofibers directly adjacent to large immune cell infiltrate clusters, revealing a link between muscle-infiltrating immune cells and myofiber mitochondrial anomalies.

## *Icos⁻/⁻* NOD muscle exhibits elevated ROS production and an oxidative stress signature

Mitochondrial alterations are typically linked to oxidative stress. Indeed, mitochondria are both a source of ROS through OXPHOS and a target of oxidative stress which damages mitochondrial components. Strikingly, we found a very high production of ROS in muscle homo-genates from *Icos⁻/⁻* NOD mice with established disease (Fig. 5a). Moreover, $H_2O_2$ production by skinned fibers from diseased mice was higher than that of *Icos⁺/⁺* NOD mice and negatively correlated with muscle weight (Fig. 5b). We corroborated the presence of oxidative stress in the muscle of mice with established myopathy by qRT-PCR analysis. Notably, we found a gradual enhancement of the expression of oxidative-response genes (*Ccs*, *Ehd2*, *Fth1*, *Park7*, *Ppp1r1b*, and *Sp1*) and antioxidant enzyme genes (*Cat*, *Gpx3*, *Gpx4*, *Gsr*, *Gss*, *Parp1*, *Prdx2*, *Prdx5*, *Srxn1*, *Txnrd1*, and *Txnrd3*) in *Icos⁻/⁻* NOD muscle, mostly reach-ing maximal levels at 35 weeks of age, and not observed in *Icos⁺/⁺* NOD mice (Fig. 5c). Oxidative stress can induce apoptosis via mitochondria-dependent and independent pathways[23]. Consistently, mRNA expres-sions of the pro-apoptotic genes *Bax*, *Casp3*, and *Casp9* were upre-gulated (Fig. 5c).

These results highlight the presence of conspicuous inflammation-driven mitochondrial defects and oxidative stress in *Icos⁻/⁻* NOD mice myositis.

## ROS-buffer therapy ameliorates myositis in *Icos⁻/⁻* NOD mice
We next aimed at dissecting the bi-directional association between mitochondrial dysfunction and inflammation by studying the effects of IFNγ-blocking and ROS-buffer therapy on *Icos⁻/⁻* NOD myositis. A pre-ventive anti-IFNγ treatment was started at week 21, since T cells were only clearly detectable in the muscle at the onset of the disease (around week 25). Treatment with anti-IFNγ abolished muscle ROS production and mitochondrial defects, restoring COX, NADH-TR, and SDH histoenzymology patterns and the percentage of COX^high fibers, and preserving mitochondrial ultrastructure (Supplementary Fig. 6a−c). Remarkably, this treatment was highly effective in pre-venting myositis, preserving locomotor activity assessed by Catwalk gait analysis (Supplementary Figs. 6d−h and 7), reducing immune

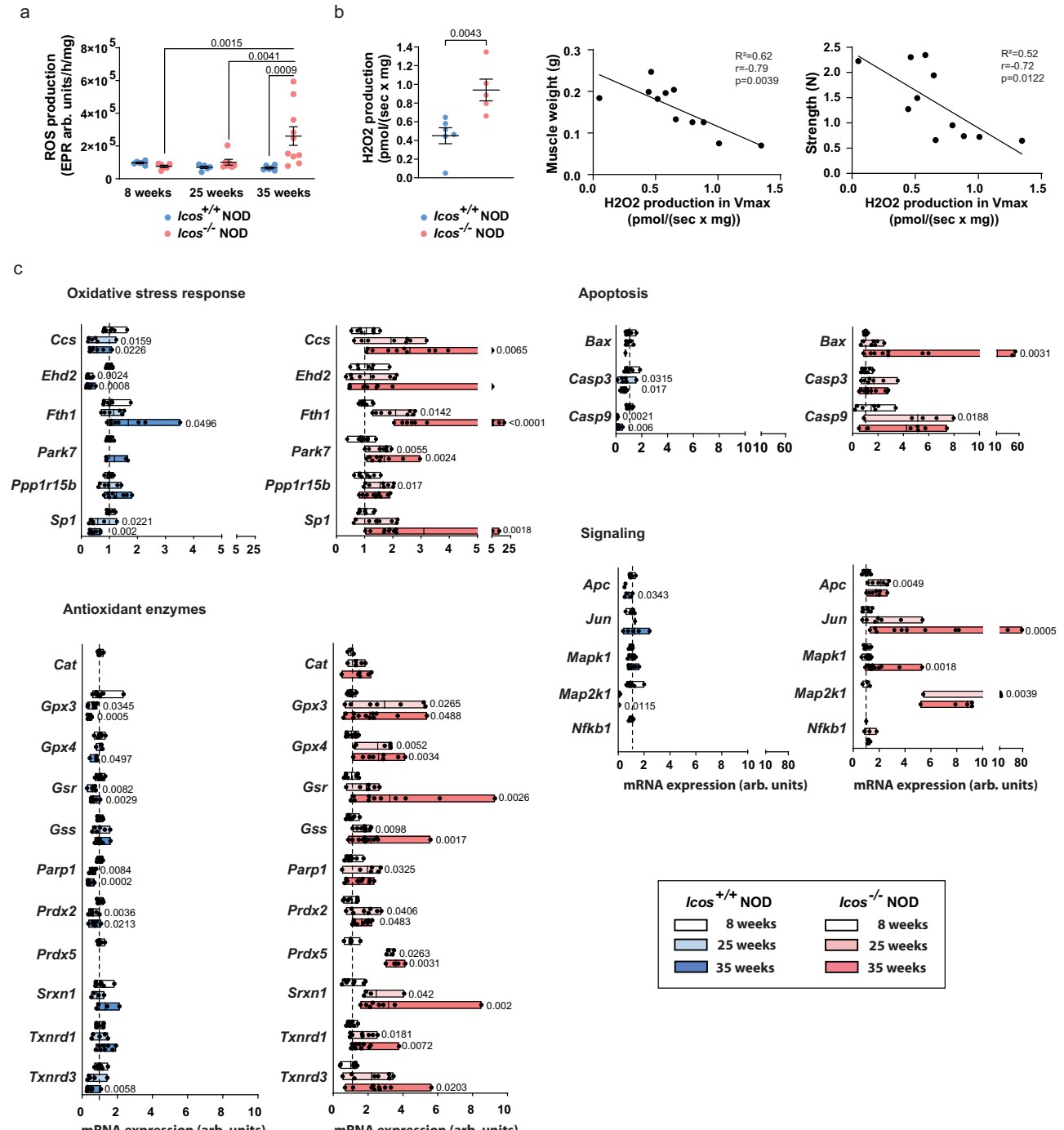

**Fig. 5 | Muscles from *Icos*<sup>-/-</sup> NOD mice exhibit enhanced ROS production and transcriptomic oxidative stress features. a** Electron paramagnetic resonance (EPR) measurement of free radical (reactive oxygen species, ROS) production in the presence of ADP in muscle homogenates from *Icos*<sup>-/-</sup> NOD and *Icos*<sup>+/+</sup> NOD mice at different time points corresponding to predisease, onset and established disease (for all *Icos*<sup>+/+</sup> NOD mice $n = 6$, for *Icos*<sup>-/-</sup> NOD mice: $n = 6$ for 8 weeks, $n = 7$ for 25 weeks and $n = 10$ for 35 weeks). **b** Ex vivo analysis of $H_2O_2$ production in *Icos*<sup>-/-</sup> NOD ($n = 5$) and *Icos*<sup>+/+</sup> NOD mice ($n = 6$) skinned muscles fibers in the presence of ADP. **c** RT-qPCR analysis targeted to oxidative stress-related genes of *Icos*<sup>-/-</sup> NOD ($n = 8$ for 8 and 25 weeks, $n = 11$ for 35 weeks) vs. *Icos*<sup>+/+</sup> NOD ($n = 8$/group for 8 and 35 weeks and $n = 6$ for 25 weeks) mice (arb. units: arbitrary units). For **a** and **b**, mean values ± s.e.m are shown. For **c**, each box expands from max to min values

and median are represented. For **a** and **b**, mean ± s.e.m and a representative experiment out of two are shown. For **a**, statistical analyses were performed using the Two-way ANOVA test with Sidak's post-hoc multicomparison. For **b**, statistical analyses were performed using the Mann–Whitney test (two-tailed) (left) and correlations by Pearson's test. For **c**, statistical analyses were performed using the Kruskal-Wallis test with uncorrected Dunn's test (for each genotype, comparison shown in the graph corresponds to comparison with values from mice at 8 weeks). *P* values displayed correspond to the comparison of 25 and 35-week-old mice data of one given genotype with 8-week-old mice data of the same genotype. For all panels, *n* correspond to the number of independent mice. Numbers on panels denote *p* values. Source data are provided as a Source Data file.

cell infiltration (Supplementary Figs. 6i and 8), the expression of IFNγ-related genes (Supplementary Fig. 6j) as well as fibrosis (Supplementary Fig. 9a).

Because of early metabolic changes in *Icos*^-/- NOD mice, we evaluated the efficacy of a preventive ROS-buffer treatment and its effect on muscle immune infiltration. Given the presence of incipient changes in metabolism-related proteins identified in the muscle proteome of these mice, we started this treatment at 14 weeks of age, way before ROS overt production. *Icos*^-/- NOD mice treated with *N*-acetyl cysteine (NAC) exhibited significantly reduced myopathy score and delayed disease onset (Fig. 6a, b). Indeed, at 23 weeks of age, when 64% of untreated *Icos*^-/- NOD mice had developed the first signs of myopathy (2.67 ± 0.51 mean score), none of the NAC-treated mice showed significant disease (1.5 ± 0.17 mean score). Muscle strength loss in *Icos*^-/- NOD mice led to severe locomotor activity impairment (Fig. 6c). Catwalk live gait monitoring revealed a decreased in most locomotor activity parameters in *Icos*^-/- NOD mice, which were significantly improved by NAC treatment (Fig. 6c). The fact that these functional tests were improved after preventive ROS-buffer treatment suggests that ROS and mitochondrial dysfunction are implicated in the muscle weakness and the effort capacity of these mice. Consistently, we observed significantly higher anterior paw grip strength in NAC-treated vs. untreated mice from 26 to 34 weeks of age (Fig. 6d). Confirming the benefit of ROS-buffer treatment on muscle activity, posterior leg muscle strength after sciatic nerve electrostimulation at the end of the study (34 weeks of age) was significantly higher in NAC vs. non-treated mice (Fig. 6e). NAC administration partially reduced muscle atrophy as evidenced by a trend to higher muscle mass than in untreated mice (Fig. 6f) and a significant reduction of fibrosis (Supplementary Fig. 9b).

Interestingly, besides restoring muscle COX, NADH-TR, and SDH histoenzymology patterns and thus mitochondrial function (Fig. 6g), and diminishing ROS levels (Fig. 6h), NAC preventive treatment also reduced inflammation as illustrated by a significantly lower degree of muscle immune cell infiltration (Fig. 6g and Supplementary Fig. 10), as well as decreased muscle expression of chemokine (*Ccl2*, *Ccl9*, and *Cxcl10*) and *Ifng* genes (Fig. 6i). In addition, this treatment reduced the levels of anti-troponin T3 antibodies (anti-TNNT3), which we had previously reported to be associated to the disease (Supplementary Table 1). This result points to a causal link between ROS/mitochondrial dysfunction and the development of inflammation.

We next evaluated the potential therapeutic effect of NAC in *Icos*^-/- NOD mice with established myositis, and thus with conspicuous mitochondrial defects and oxidative stress, by starting a 5-week NAC treatment when the clinical score reached 2. This regimen halted disease progression, improving locomotor activity (Supplementary Fig. 11a, b). Supporting an amelioration of muscle disease, both grip strength and muscle strength after sciatic nerve electrostimulation were improved by 5 weeks of NAC treatment (Supplementary Fig. 11c, d). Fibrosis was significantly reduced by this treatment (Supplementary Fig. 9c). Moreover, COX, NADH-TR, and SDH patterns were partially restored, and a trend to a decrease in ROS production was found (Supplementary Fig. 11e, f). However, in this therapeutic setting, NAC had a lesser impact on immune cell infiltration and chemokine/cytokine production than in the preventive setting, with only a non-significant trend to reduced *Ifng* gene expression (Supplementary Figs. 11e, g and 12), and no impact on anti-TNNT3 antibody levels (Supplementary Table 1).

Regarding mitochondrial respiration, the RCR was restored by NAC treatment, indicating an amelioration of mitochondrial function (Supplementary Fig. 13a). Consistently, NAC-treated mice had less abnormal mitochondria than untreated mice (Supplementary Fig. 13b).

These results demonstrate the important pathological role of oxidative stress in myositis and suggest the presence of a crosstalk between immune cells and muscle mitochondria.

## An IFNγ signature in myositis patients is associated with mitochondrial dysfunction and oxidative stress, and IFNγ impairs mitochondria in myoblasts in vitro

In order to confirm the association between IFNγ and mitochondrial abnormalities observed in mice, we analyzed bulk RNAseq data from muscle biopsies from 44 DM patients who tested positive for myositis-specific autoantibodies directed against NXP2 (n = 14), TIF1γ (n = 12), Mi2 (n = 12) or MDA5 (n = 6), compared to 33 histologically normal muscle biopsies (NT). The expression of multiple respiratory chain genes encoded by mitochondrial DNA (mtDNA) was reduced in DM patients compared to histologically normal controls (Fig. 7a and Supplementary Table 2). There was a negative correlation between the expression of OXPHOS and IFNγ-stimulated genes (*GBP2*, *IFI30*, and *IFNG* itself), with the lowest mitochondrial gene expression in muscle biopsies with a prominent IFNγ-induced gene signature (Fig. 7b and Supplementary Table 3). *GBP* expression levels were higher in DM than in NT samples, with the highest levels being associated with the lowest expression of mitochondrial OXPHOS genes (Supplementary Fig. 14). Expression of mitochondrial genes also negatively correlated with the expression of canonical immune cell markers (Fig. 7b and Supplementary Table 3). As expected, we found a positive correlation between mitochondrial and skeletal muscle contraction machinery genes, supporting the association between muscle health and mitochondrial function (Fig. 7b and Supplementary Table 3). We found no relevant changes in mitochondrial gene expression among serologic groups, indicating that the metabolic defects observed is a general pathophysiological mechanism of the disease rather than ascribed to a specific myositis subset (Supplementary Fig. 15). No differences were found when stratified by sex (Supplementary Fig. 16).

To further support the causative role of IFNγ in the development of mitochondrial anomalies, we investigated whether IFNγ would affect mitochondrial gene expression on differentiating human myoblasts. IFNγ triggered the IFNγ-inducible genes *GFP2* and *IFI30*, indicating the responsiveness of myoblasts to this cytokine (Fig. 7c). Importantly, IFNγ led to a remarkable downregulation of the same OXPHOS mitochondrial genes that were reduced in our bulk transcriptome analysis of DM patient biopsies (Fig. 7c). Moreover, electron microscopy of IFNγ-treated myotubes showed the presence of mitochondrial ultrastructural abnormalities, such as enlargement and signs of mitophagy, which were prevented by treatment with a JAK inhibitor (ruxolitinib or baricitinib) (Fig. 7d).

These therefore confirm that IFNγ exerts a negative effect on human muscle and a pathogenic role in myositis as in mice.

## Discussion

Oxidative stress has been linked to the pathogenesis of multiple inflammatory and autoimmune diseases including rheumatologic systemic diseases (systemic lupus erythematosus, systemic sclerosis, rheumatoid arthritis), inflammatory bowel diseases (Crohn's disease, ulcerative colitis), neurologic disorders (multiple sclerosis), skin diseases (scleroderma) and autoimmune endothelial pathology[24–26]. Attenuating oxidative stress directly with NAC, a widely known ROS-buffer compound with very few adverse effects, ameliorated myositis in *Icos*^-/- NOD mice, confirming its pathogenic role. Expectedly, treated mice did not develop diabetes (Supplementary Table 4).

Several mechanisms may explain how inflammation and oxidative stress fuel each other and contribute to disease development and/or progression. High ROS levels oxidize macromolecules, causing structural modifications and cell death leading to inflammation. In addition to losing their functionality, some of these modified molecules may become neoantigens endorsing autoimmunity[27–29]. For example, high serum levels of oxidized proteins and lipids, autoantibodies against lipids and ROS scavengers, and oxidized lipid immune complexes in SLE patients correlate with active disease[30–34]. Other evidence suggests

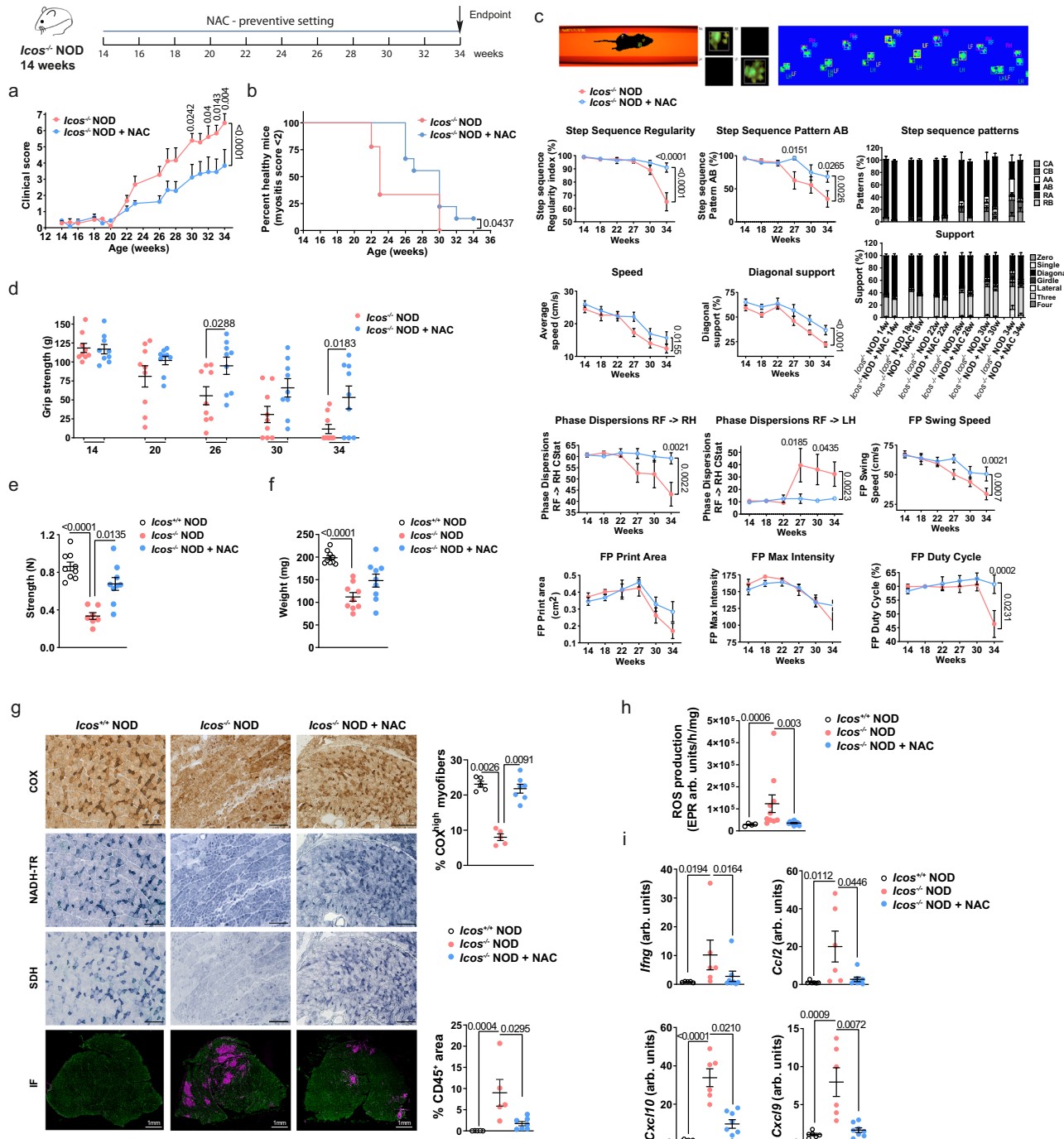

**Fig. 6 | Preventive NAC administration alleviates *Icos*⁻/⁻ NOD mice myositis.** *Icos*⁻/⁻ NOD mice were treated with NAC (2 g/L) from 14 weeks of age until 34 weeks of age (*n* = 10 mice/group). **a** Clinical score. **b** Percentage of disease-free mice. **c** Locomotor activity (Catwalk XT). **d** Grip strength. **e** Muscle strength after sciatic nerve stimulation. **f** Muscle weight. **g** Histopathological analysis. COX, NADH-TR, and SDH representative histoenzymology stainings (scale bars, 200 μm) and CD45/laminin immunofluorescence staining (scale bars, 1 mm). Graphs (right) depict the percentage of COX^high fibers and the mean CD45-positively stained area with respect to total muscle area. *Icos*⁺/⁺ NOD (*n* = 5), *Icos*⁻/⁻ NOD (*n* = 5), *Icos*⁻/⁻ NOD + NAC (*n* = 7). **h** EPR measurement of ROS production (*Icos*⁺/⁺ NOD (*n* = 4), *Icos*⁻/⁻ NOD (*n* = 10), *Icos*⁻/⁻ NOD + NAC (*n* = 10)). **i** Chemokine and cytokine mRNA expressions

(arb. units: arbitrary units). *Icos*⁺/⁺ NOD (*n* = 6), *Icos*⁻/⁻ NOD (*n* = 6), *Icos*⁻/⁻ NOD + NAC (*n* = 8). For **a** and **c**, statistical analysis was performed using Two-way ANOVA and Sidak's multiple comparison post-hoc test. For **b**, Log-rank (Mantel Cox) test was used. For **d**, statistical analyses were performed using Two-way ANOVA with Sidak's post-hoc test and for **e–i**, statistical analyses were performed using the Kruskal–Wallis test with uncorrected Dunn's multiple comparison test. For all panels, *n* values correspond to the number of independent mice. Data correspond to the mean values ± s.e.m and numbers on panels denote *p* values. For all, a representative experiment out of two is shown. Source data are provided as a Source Data file.

a subtler modulation of inflammatory responses by ROS, for example through upregulation of TLR trafficking to the cell membrane[35]. Along those lines, NAC inhibition of the NF-κB pathway, which has a key role in inflammation, has been suggested by in vitro studies but has nevertheless remained controversial[20,36,37]. Conversely, as one example of inflammation-triggered oxidative stress, TLR engagement can lead to ROS formation through the activation of NADPH oxidases and the upregulation of inducible nitric oxide synthase (iNOS)[38]. In turn,

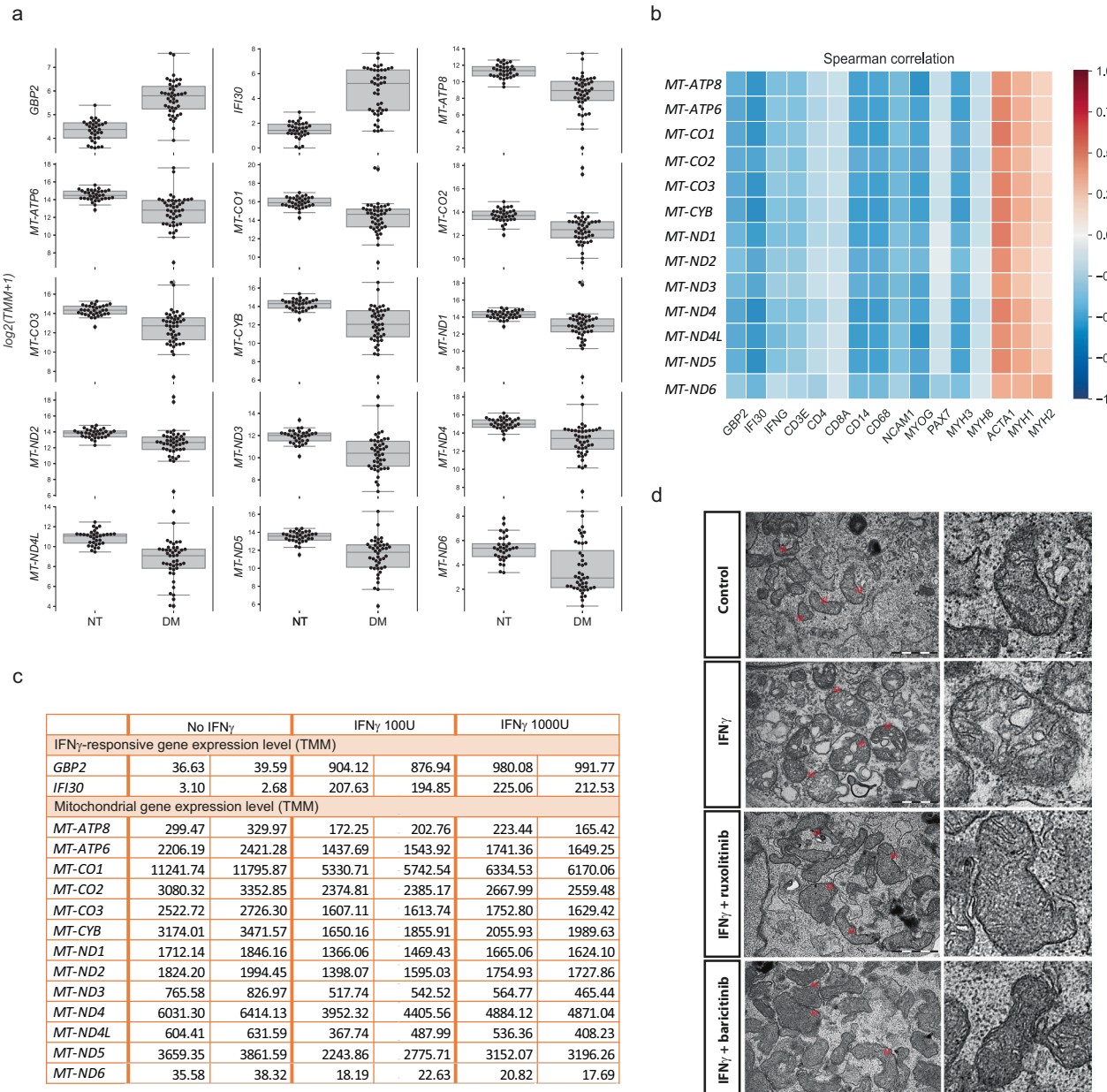

**Fig. 7 | Bulk transcriptome analysis of dermatomyositis muscle biopsies and human myoblast culture with IFNγ suggest a causative link between this cytokine and mitochondrial dysfunction. a** Analysis of bulk transcriptomic data of muscle biopsies from 44 DM patients tested positive for myositis-specific autoantibodies against NXP2 (*n* = 14), TIF1γ (*n* = 12), Mi2 (*n* = 12), and MDA5 (*n* = 6), and from 33 histologically normal muscle biopsies focusing on mitochondrial genes (*ATP8, ATP6, CO1, CO2, CO3, CYB, ND1, ND2, ND3, ND4, ND4L, ND5,* and *ND6*). NT normal tissue, DM dermatomyositis. *n* corresponds to independent subjects. Box plots bounds to 25th to 75th percentiles, with line at the median, and whiskers correspond to 1.5 times the interquantile range (1.5×[Q3-Q1]). **b** Correlation of the expression of these mitochondrial genes with IFNγ-induced genes (*GBP2, IFI30, IFNG*) and other genes related to myositis disease activity: immune (*CD3E, CD4,*

*CD8A, CD14, CD68, NCAM1*), myofiber regenerative (*MYOG, PAX7, MYH3, MYH8*) and mature myofiber markers (*ACTA1, MYH1, MYH2*). **c** Effect of IFNγ (100 U/mL or 1000 U/mL) on mitochondrial gene expression in differentiating human skeletal muscle myoblasts (*n* = 2 wells/condition). Expression Trimmed Means of M values (TMM) of *GBP2, IFI30, ATP8, ATP6, CO1, CO2, CO3, CYB, ND1, ND2, ND3, ND4, ND4L, ND5,* and *ND6*. **d** Electron microscopy of myotubes treated with IFNγ and IFNγ + JAK inhibitors (ruxolitinib 1 μM or baricitinib 1 μM) (left, scale bar 1 μm). A zoomed insert on a representative mitochondrion is shown on the right (scale bar 0.2 μm). Images are representative of *n* = 3 wells/condition. For **a**, the Benjamini–Hochberg was used to adjust for multiple comparisons and *p* values are included in Supplementary Table 2. Source data are provided as a Source Data file.

---

inflammation-driven oxidative stress triggers protective mechanisms to dampen its deleterious effects. For example, IFN-induced immunoproteasomes protect cell viability under oxidative stress conditions[39]. Nevertheless, these may be insufficient to counteract extensive damage and a vicious circle between inflammation and oxidative stress may then contribute to the maintenance of chronic inflammation.

Mitochondrial pathology is closely linked to oxidative stress and has been reported in IIMs[40] and other inflammatory diseases[41–43]. Spatial transcriptomics revealed a mitochondrial damage gene signature in myofibers from *Icos*[-/-] NOD muscles, which was accentuated when located in contact with immune cell infiltrates, supporting a cross-talk between inflammatory cells and muscle mitochondria. IFNs, widely implicated in human IIMs, and particularly IFNγ, also involved in

*Icos*[-/-] NOD mice myositis, appear to play a central role in this process. Our previous findings using DM muscle samples, cell and muscle models identified mitochondrial alterations in perifascicular myofibers induced by IFNβ, and that mitochondrial dysfunction contributes to both exercise limitation and maintenance of inflammation in the muscle[15]. Here, we further revealed a correlation between IFNγ and diminished mitochondrial gene expression in DM patients, consistent with the deleterious effects of proinflammatory cytokines on mitochondria reported in some other cell types[44–46]. Furthermore, we found that exposure of differentiating human myoblasts to IFNγ reduced the expression of OXPHOS genes and altered mitochondrial ultrastructure, confirming IFNγ-driven mitochondrial alterations in myositis. In this line, *Icos*[-/-] NOD mice treatment with an IFNγ-blocking antibody ameliorated the disease, abolishing not only inflammation but also oxidative stress and mitochondrial anomalies. Interestingly, it has been recently reported that administration of IFNγ to C57BL/6 mice limits the beneficial effects of exercise by impairing muscle mitochondrial function[47].

Skeletal muscle contraction demands high energy levels and thus greatly relies on optimal mitochondrial function as a major source of ATP[48]. Given their critical role as bioenergetic powerhouses, it is not surprising here that mitochondrial defects led to a profound metabolic imbalance contributing to myositis pathogenesis. Mitochondrial anomalies in *Icos*[-/-] NOD mice with myositis could be both a cause and a consequence of oxidative stress. Mitochondria are an important source of ROS (mtROS) through OXPHOS, especially when malfunctioning[49–51]. Toxic ROS levels can result in mitochondrial membrane and mtDNA damage and electron transport chain dysfunction, amplifying oxidative stress and leading to cell death[52]. In addition, severe energy or oxidative stress converts mitochondria to a dysfunctional metabolic state associated with pathological swelling[53]. Proteome, spatial transcriptome, and electron microscopy data in this study support the presence of abnormal mitochondria in the *Icos*[-/-] NOD mice model.

Growing evidence suggest that mitochondria can regulate innate and adaptive immunity[49,54]. For instance, mitochondria modulate pattern-recognition receptor signal transduction. This is the case of cytosolic retinoic acid-inducible gene I (RIG-1)-like receptors (RLRs), which once stimulated by viral dsRNA interact with the antiviral signalling protein (MAVS) at the outer mitochondrial membrane, leading to NF-kB activation and engagement of antiviral responses[49,55]. Evidence also supports that mitochondria modulate TLR and NLRP3 inflammasome activation[49,56,57]. To note, RIG-1 and TLRs expressions are elevated in DM patient muscles[58–60]. In addition, mitochondrial molecules released into the cytoplasm or the extracellular *milieu* upon damage or stress such as mtDNA, ATP, cardiolipin, and formyl peptides, act as danger-associated molecular patterns (DAMPs) causing inflammation[49,61]. Moreover, it has been recently demonstrated that the loss of mitochondrial membrane integrity initiates NF-kB signalling[62]. Hence, mitochondrial anomalies in *Icos*[-/-] NOD mice contribute to the disease by causing energy failure, enhanced ROS production, and worsening immune dysregulation.

Multiple evidence supports that metabolism depends on the coordinated action of mitochondria and peroxisomes[63]. Interestingly, we identified the presence of a downregulation of certain peroxisomal gene expression and protein levels in *Icos*[-/-] NOD mice (Supplementary Fig. 17). Consistently, gene expressions of two peroxisomal genes downregulated in mice, *Phyh* and *Ech1*, were also significantly downregulated in human DM biopsies (Supplementary Fig. 17).

Although mitochondrial impairment and oxidative stress were manifest in the muscle of mice with established myositis, our proteome analysis highlighted that certain alterations in the mitochondrial electron transport and other metabolic pathways were already present in asymptomatic *Icos*[-/-] NOD mice, when muscle inflammation and histopathology traits are not yet overt. Interestingly, mice with

conditional syngeneic class I MHC (H-2Kb) skeletal muscle overexpression, who develop spontaneous myositis, exhibit a deficiency in purine metabolism prior to immune cell infiltration[64]. Early electron transport chain disturbances in our mice may fuel the initial inflammatory response in the muscle of *Icos*[-/-] NOD mice. Therefore, whereas the current view is that muscle inflammation causes myofiber necrosis and subsequent regeneration through cell- and cytokine-mediated cytotoxicity, the present results add mitochondrial defects and oxidative stress as a major pathogenic component to the mechanisms of myositis, being both potential triggers as well as a consequence of muscle dysregulated inflammation.

This murine model presents some limitations. Certainly, it has the advantage of being spontaneous and not antigen-biased as compared to immunization models in which we also previously observed some oxidative stress and reduced mitochondrial OXPHOS[15]. Yet, myositis in *Icos*[-/-] NOD mice does not completely recapitulate all aspects of the human disease. While human IIMs are characterized by the presence of a series of autoantibodies with a key diagnostic value, we could not detect the myositis-specific antibodies found in patients in these mice. Also, we could not find skin disease in these mice and therefore assess the extramuscular signs otherwise observed in myositis such as in dermatomyositis.

In conclusion, our study using a murine model of myositis suggests the presence of a self-sustaining loop between inflammation and mitochondrial dysfunction/oxidative stress in the pathogenesis of myositis, and provide evidence supporting its relevance in human myositis, pointing at mitochondria as a possible new therapeutic target.

## Methods

### Inclusion and ethics statement
All collaborators of this study have fulfilled the criteria for authorship required by Nature Portfolio journals, as their participation was essential for the design and execution of the study. Roles and responsibilities were agreed ahead of the research among collaborators.

### Mice
*Icos*[+/+] NOD and *Icos*[-/-] NOD mice[65] (*Mus Musculus*), were bred and housed in our facilities under specific pathogen-free (SPF-free) conditions. Female mice ranging from 8-35 weeks of age were used in our studies. Experimental/control mice were housed separately. Mice were housed under a 12 h light/dark cycle with food pellets and drinking water provided *ad libitum* and used for the studies. Room temperature was maintained within the range of 21–23 °C and relative humidity ranged between 45-55%. All protocols were performed with the agreement of the local animal ethics committee (*Comité d'éthique NOrmand en Matière d'EXpérimentation Animale*) and of the ministerial committee for animal experimentation (APAFIS#8780). Animal care and experimentation complied with the European directive for the use and care of laboratory animals (2010/63/UE). Diabetes was systematically assessed by the use of urine sticks, and only non-diabetic mice were used in this study. For ROS-buffer treatment studies, a group of mice received N-acetyl cysteine (NAC, Sigma) in the drinking water (2 g/L), with the solution being changed twice a week, starting in animals at 14 weeks of age (preventive setting) until 32 weeks of age or starting in mice at onset (clinical score >2) of the disease (therapeutic setting) for five weeks. We chose to use only females because the incidence of spontaneous myopathy is significantly higher (70%) than in males (20%) as we reported previously[65]. Mice were euthanised by cervical dislocation.

### Assessment of myositis severity
The degree of myopathy was quantified using a clinical score from 0 to 10 as previously described[66]. Briefly, this consisted in a cumulative

score based on the sum of two separate scores (from 0 to 3) given to front and hind paws based on the degree of spontaneous limb flexion and impairment of march on a grid, a global score (from 0 to 3) based on hunch position when holding by tail suspension, and an additional point (+1) given for abnormal breathing. In parallel, front paw grip test performance was used to evaluate muscle strength (Bioseb). Muscle strength was also determined at the experiment end point by directly measuring muscle contraction upon sciatic nerve electrostimulation on mice under ketamine/xylazine anesthesia. In brief, the distal tendon of the gastrocnemius was detached and tied to an isometric transducer (PowerLab Station, AD Instruments). The sciatic nerve was stimulated distally (2 V, 70 Hz, 300 ms), the response to stimulation was recorded and maximal force was determined. Measurement and analysis were performed on LabChart7 software (AD Instruments), as adapted from[67].

In parallel, a Catwalk apparatus (Noldus Information Technology) was used to evaluate locomotor activity. Three compliant trials were recorded for each animal. The experiment was performed in a darkened room and sheltered from noise. Analysis was performed using the Catwalk XT 10.5 Software.

At the end of the experiment, quadriceps muscles were immediately frozen in isopentane for further histological analyses or in liquid nitrogen for reactive oxygen species (ROS), proteome, and transcriptome analyses, and stored at −80 °C.

## Sample preparation for LC-MS/MS

Dissected muscles were grinded using FastPrep® technology (MP Biomedicals). Seven hundred μL of solubilization buffer (Urea 7 M, Thiourea 2 M, DTT 20 mM, CHAPS 2%, C7BzO 0.5%, and tributylphosphine 5 μL) were added to each sample in lysing matrix D (Biorad). Homogenization was performed using the following parameters: 4 cycles (40 s) and speed at 6 m/s. Supernatants were transferred to a clean tube after centrifugation (11,000×$g$, 7 min) and stored at −80 °C. Bradford assay was performed to assess protein concentration according to the manufacturer's instructions (Biorad). Absorbance at 595 nm was measured with the Wallac 1420 Victor3™ (Perkin Elmer). Peptide fraction was obtained to perform bottom-up analysis. For each extract, 25 μg of proteins were loaded into a 7% polyacrylamide gel (acrylamide/bis-acrylamide 30% [29/1], Sigma-Aldrich) and an electrophoresis was performed (90 min, 10–20 mA) to stack all proteins in a small piece of gel. After Coomassie blue staining, protein bands were excised and immersed in a reductive buffer (5 mM DTT), then in an alkylated buffer (20 mM iodoacetamide). After a washing step using acetonitrile, gel bands were submitted to protein digestion by 1 μg of trypsin (Promega). After overnight incubation at 37 °C, several steps of peptide extraction were performed using acetonitrile and 0.1% formic acid (FA) in water. Finally, the peptide fraction was dried.

## Nano LC-MS/MS analysis

For each sample ($n = 30$), the peptide fraction was solubilized in 0.1% FA to obtain a final concentration of 0.2 μg/μL. Then, 1 μL was analyzed on a Q-exactive Plus Orbitrap™ Mass Spectrometer (Thermo Scientific) equipped with a nanoESI source and coupled to a chromatographic system, RSLCnano Ultimate 3000 (Thermo Scientific). The separation was carried out using an EASY Spray column (50 cm, 3 μm, 0.075 mm, 100 A, flow rate 300 nL/min, Thermo Scientific). The mobile phase was composed of buffer A (H2O/FA [100/0.1]) and buffer B (acetonitrile/H2O/FA [80/20/0.1]). The elution gradient duration was 180 min: 0–2 min, 2% B; 2–130 min, 2–40% B; 130–140 min, 40–95% B; 140–150 min, 95% B; 150–180 min, 2% B. The mass spectrometer was operated in positive mode with HCD fragmentation (capillary voltage 1.9 kV, capillary temperature 275 °C, and m/z detection range 400-1800). The resolution was 70,000 in MS1 and 17,500 in MS2. The 10 most intense peptide ions were selected and the fragmentation

occurred with a normalized collision energy of 28. Dynamic exclusion of already fragmented precursor ions was applied for 20 s.

After MS analysis, raw data were imported in Progenesis LC-MS software (Non-Linear Dynamics, version 4.1). For comparison, one sample was set as a reference and the retention time of all other samples within the experiment were aligned. After alignment and normalization, statistical analysis was performed for one-way ANOVA calculations. For quantification, peptide features presenting a $p$-value less than 0.05 were retained. MS2 spectra from selected peptides were exported for peptide identification with Mascot (Matrix Science). Database searches were performed with the following parameter: *mus musculus* taxonomy (16951 sequences); 1 missed cleavage; variable modifications (carbamidomethyl of cysteine and oxidation of methionine). Mass tolerances for precursor and fragment ions were 5 ppm and 0.02 Da respectively. False discovery rates (FDR) were calculated using a decoy-fusion approach in Mascot. Identified spectrum matches with −10logP value of 17 or higher were kept, at an FDR threshold of 3%. Mascot search results were imported into Progenesis LC-MS. For each condition, the total cumulative abundance of protein was calculated by summing the abundances of peptides and used to perform statistical analysis ('one-way ANOVA' Progenesis statistical box). Proteins identified with less than 2 peptides were discarded from further analysis. All of the dysregulated proteins were integrated as inputs for functional analysis (STRING https://string-db.org/; Ingenuity Pathway Analysis (IPA), Qiagen).

## Nanostring spatial transcriptomic analysis

Six-micrometer-thick cryosections were prepared using the GeoMx Digital Spatial Profiler (DSP) Whole Transcriptome Atlas (WTA) slide prep protocol from NanoString Technologies. Mouse WTA probes targeting more than 19,000 targets were hybridized, and after two washes with 2X SSC, slides were loaded on the GeoMx DSP. Entire slides with *Icos*+/+ NOD and *Icos*-/- NOD sections were imaged at ×20 magnification, and Regions of Interest (ROIs) (4 *Icos*+/+ NOD and 14 *Icos*-/- NOD) were selected. The GeoMx DSP segmentation tool allowed the separation of different areas of illumination (AOI) on the basis of morphology markers (CD45 and desmin). This strategy allowed the analysis of RNA expression from myofibers (desmin+CD45-) from *Icos*+/+ NOD ($n = 4$) and from *Icos*-/- NOD mice, the latter being non-adjacent (designated as "*Icos*-/- NOD mice PROX") ($n = 6$) or adjacent (designated as "*Icos*-/- NOD mice ADJ") ($n = 8$) to immune cell infiltrate clusters (desmin-CD45+). The GeoMx exposed AOIs to ultraviolet light (385 nm), releasing the indexing oligos and collecting them with a microcapillary. Indexing oligos were then deposited in a 96-well plate, dried down overnight, and resuspended in 10 μl of RNAse-free water.

PCR was used to generate sequencing libraries from the photoreleased indexing oligos and ROI-specific Illumina adapter sequences, and unique i5 and i7 sample indices were added. Each PCR reaction contained 4 μl of indexing oligos, 4 μl of indexing primer mix, and 2 μl of NanoString 5× PCR Master, and PCRs were pooled and purified twice using AMPure XP beads according to the manufacturer's protocol (Beckman Coulter, A63881). Pooled libraries were sequenced at 2 × 27 base pairs and with the dual-indexing workflow on an Illumina NovaSeq. Data was analyzed using the GeoMx DSP Suite Software. Q3 normalization of data using the top 25% of expressers to normalize across segments was performed.

## qRT-PCR analysis

RNA was isolated from snap-frozen muscle tissues by using Tri-reagent (Sigma) according to the manufacturer's instructions. RNA quality and quantification were determined with the Agilent RNA 6000 Nano Kit and the Agilent 2100 bioanalyzer. One μg of RNA was retrotranscribed using the ImPromII reverse transcriptase kit (Promega).

For the oxidative stress panel, real-time PCR experiments were performed and monitored by ABI Prism 7500 Sequence Detection

System (Life Technologies). *Rplp0*, *Tfrc*, *Hsp90ab1*, and *Nono* were used as housekeeping genes.

For quantification of *Ifng*, *Ifnb*, *Ccl2*, *Cxcl9*, *Cxcl10* and *Gbp2*, amplification was performed with the LightCycler 480 System SW 1.5.1 (Roche). *Hsp90ab1* was used as housekeeping gene.

Primer sequences are included in Supplementary Table 5.

The relative expression between a given sample and a reference sample was calculated using the $2^{-\Delta\Delta Ct}$ algorithm, where ΔCt is the difference in the Ct values for the target gene and the average of the four housekeeping genes (oxidative stress panel analysis) or *Hsp90ab1* (*Ifng*, *Ifnb*, *Ccl2*, *Cxcl9*, *Cxcl10*, *Gbp2* analysis), and ΔΔCt is the difference between each sample and the basal reference value.

### Electron paramagnetic resonance spectroscopy

Muscles were frozen in liquid nitrogen and kept at −80 °C until free radical production measurement by electron paramagnetic resonance (EPR) spectroscopy using the spin probe 1-hydroxy-3-methoxycarbonyl-2,2,5,5-tetramethyl pyrrolidine hydro-chloride (CMH; Noxygen). Tissues were homogenized in Krebs-HEPES buffer (pH7.4) and then were incubated in Krebs-HEPES buffer containing diethyldithiocarbamic acid silver salt (5 mM), deferoxamine (25 mM), and CMH (50 mM) in 24-well plates at 37 °C for 1 h. Samples were immediately frozen in liquid nitrogen to stop the reaction. Wells without tissue were used as blank. Oxidized CMH spectra were recorded with a MiniScope MS-200 (Magnettech). EPR acquisition parameters were as follows: Bo-field 3325.96 G; microwave power 1 mW; modulation amplitude 5 G; sweep time 60 sec. Spectra intensity was expressed in arbitrary units, normalized per mg of protein.

### Mitochondrial respiration and $H_2O_2$ production assessment

Mitochondrial function was assessed in situ by measuring muscle oxygen consumption and $H_2O_2$ production in saponin-skinned muscle fibers using high-resolution oxygraphy (O2K, Oroboros instruments) as previously described[15,68] and as below.

Basal oxygen consumption (V0) due to proton leak in the presence of glutamate (10 mM) and malate (2 mM) was first measured. Maximal fiber respiration (Vmax) was recorded after addition of saturating amounts of ADP (2.5 mM) leading to electron flow through complexes I, III, and IV and oxidative phosphorylation of ADP. The respiratory control ratio (RCR) was computed as the ratio between V0, providing a measure of the coupling (efficiency) of mitochondrial oxidation to ATP production. At the same time, $H_2O_2$ production was measured with Amplex Red (Invitrogen, 0.02 mM) and horseradish peroxidase (Invitrogen, 1 U/mL) in the presence of ADP or succinate; fluorescence was measured continuously at 587 nm. Respiratory control ratio (RCR) was computed as the ratio between Vmax and V0.

### Muscle histological analyses

For all histological studies, quadriceps muscles were frozen in cold isopentane, and 7-µm cross-sections were prepared with a cryostat. Histoenzymology stainings of COX (Cytochrome c oxidase), NADH-TR (nicotinamide adenine dinucleotide tetrazolium reductase), and SDH (succinate dehydrogenase) activities were performed according to standard procedures. Briefly, for COX activity, sections were incubated with 0.08 mM cytochrome c (Sigma), 0.2 M sucrose (Sigma), a few crystals of catalase (Sigma), and 5 mM DAB (Sigma) in 0.2 M phosphate buffer for 2 h at 37 °C. For NADH-TR staining, muscle sections were incubated in 1.2 mM NADH (Sigma) and 1.2 mM nitro-blue tetrazolium (NBT; Sigma) in 0.05 M Tris buffer for 40 min at 37 °C. For SDH staining, sections were incubated in 170 mM sodium succinate salt (Sigma) and 1.2 mM NBT in 0.2 M phosphate buffer for 1 h at 37 °C. Whole muscle cross-sections (two sections per animal) were imaged in their entirety at 10X magnification using a Zeiss Axioscope 7 microscope and Zen software. Fibrosis was revealed by Sirius red staining according to standard protocols as in ref. 19.

Immunofluorescence staining was performed by incubating muscle sections with primary anti-CD45 (1:300, Sony) and anti-laminin (1:400, Invitrogen) antibodies overnight at 4 °C. Sections were then incubated with Alexa Fluor 488 or Cy5-conjugated secondary antibodies (1:400, Jackson ImmunoResearch) for 1 h at room temperature. Antibody catalog numbers and dilutions are included in Supplementary Table 6.

Slides were mounted using Fluoromount-G (Southern Biotech). Whole muscle section images (2 sections per animal) were obtained with the Leica Thunder Tissue Imager system and LAS Software at 10X magnification.

Quantifications of the percentage of Sirius red positive and immunoreactive areas on whole muscle sections was determined using Fiji software[69].

### Electron microscopy

For muscle mitochondria studies, muscle samples were fixed overnight with 2.5% glutaraldehyde (Sigma-Aldrich) and 2% paraformaldehyde in 0.1 M cacodylate buffer (pH 7.3). Post-fixation was done with 1% osmium tetroxide for 2 h at 4 °C (Merck). After a gradual ethanol dehydration, tissues were infiltrated with Spurr resin. Polymerized resin blocks were sectioned using a Leica UC7 microtome (Leica Microsystems). Ninety nm sections were collected on Formvar-coated slot grids (Electron Microscopy Sciences) and contrasted with 0.5% of uranyl acetate and Reynolds lead citrate. Sections were observed with a Tecnai 12 microscope operated at 80 kV, and images of 10-15 different areas from 15-20 muscle fibers from each mouse were taken at ×8200 or ×26,500 magnification.

Regarding electron microscopy on human myotubes (LHCN-M2), cells were fixed by immersion in 2.5% glutaraldehyde and 2.5% paraformaldehyde in cacodylate buffer (0.1 M, pH 7.4). The samples were postfixed in 1% osmium tetroxide in 0.1 M cacodylate buffer for 1 h at 4 °C. After a gradual ethanol dehydration, samples were embedded in Epon 812. Ultrathin sections were cut at 70 nm (Leica Ultracut UCT), contrasted with uranyl acetate and lead citrate and examined at 70kv with a Morgagni 268D electron microscope (FEI Electron Optics, Eindhoven, the Netherlands). Images were captured digitally by Mega View III camera (Soft Imaging System).

### Patients

All muscle biopsies analyzed in this study belonged to participants in institutional review board (IRB)-approved longitudinal cohorts from the National Institutes of Health in Bethesda, MD the Johns Hopkins Myositis Center in Baltimore, MD, the Vall d'Hebron Hospital, and the Clinic Hospital in Barcelona[70]. All participants signed a written informed consent form, and all methods were performed in accordance with the relevant guidelines and regulations. Patients met the ACR criteria for DM[71] and tested positive for one of the following myositis-specific autoantibodies (MSA): anti-NXP2, anti-Mi2, anti-TIF1g or anti-MDA5. ELISA, immunoprecipitation of proteins produced by in vitro transcription and translation (IVTT-IP), line blotting (EUROLINE myositis profile), and immunoprecipitation from 35S-methionine-labeled HeLa cell lysates were used to test for autoantibodies. All dermatomyositis cases were adults. Histologically normal muscle biopsies were obtained from the National Institutes of Health ($n = 13$), the University of Kentucky Skeletal Muscle Biobank ($n = 8$), and the Johns Hopkins Neuromuscular Pathology Laboratory ($n = 12$).

### RNA sequencing

Bulk RNAseq of frozen muscle biopsy samples was carried out as previously described[70,72]. TRIzol (Thermo Fisher Scientific) was used for RNA extraction. Libraries were either created using the NEBNext Poly(A) mRNA Magnetic Isolation Module and UltraTM II Directional RNA Library Prep Kit for Illumina (New England BioLabs) or the

NeoPrep system in accordance with the TruSeqM Stranded mRNA Library Prep protocol (Illumina).

## Culture of differentiating human skeletal muscle myoblasts and treatment with IFNγ

Normal human skeletal muscle myoblasts (Lonza) were cultured as previously described[70]. When reaching 80% confluence, myotube differentiation was induced by replacement of the growth medium by differentiation medium (Dulbecco's modified Eagle's medium with 2% horse serum and L-glutamine). To examine the effect of IFNγ on mitochondrial gene expression, Myoblasts were treated daily with 100 U/mL or 1000 U/mL of recombinant human IFNγ (PeproTech) for 7 days. On the last day of treatment, cells were harvested for RNA extraction and RNA sequencing. For electron microscopy studies, LHCN-M2 myoblasts were differentiated into myotubes for 5 days and then treated with IFNγ 1000 U/ml with or without ruxolitinib (10 μM) ou baricitinib (10 μM) for 6 days.

## Addressable laser bead immunoassay for anti-troponin T3

OriGene Technologies provided the MYC/DDK-tagged troponin T3 (TNNT3) protein. Using a Bio-Plex amine coupling kit (Bio-Rad), 10 μg of recombinant proteins were linked to Bio-Plex Pro magnetic carboxylated beads. Using a commercial antibody that targets the TNNT3 protein with anti-DDK from OriGene Technologies, the tests were validated.

Beads were incubated for two hours at room temperature with diluted sera or control antibodies. Following washing, 50 μL of streptavidin R phycoerythrin (Bio-Rad) was incubated for 15 min, after which a secondary biotinylated antibody (SouthernBiotech) was added and incubated for one hour. Each sample's mean fluorescence intensity (MFI) was calculated using a Bio-Plex device (Bio-Rad).

## Statistical and bioinformatic analysis

Statistical tests (log rank, two-tailed Mann–Whitney, Kruskal–Wallis, and one-way or two-way ANOVA with Sidak's *post-hoc* multicomparison) were performed as indicated in the figure legends using GraphPad Prism Version 8.0 software (GraphPad Software, La Jolla, CA, USA). For proteome analysis, statistical analysis was performed using the inbuilt Progenesis LC-MS software (Non-Linear Dynamics, v4.1) statistical box called 'one-way ANOVA'. For IPA pathway analysis, the Fisher's Exact Test was used to calculate the statistical significance of overlap of dataset molecules with various sets of molecules that represent annotations such as Canonical Pathways. For Nanostring data analysis, a Linear Mixed Model (LMM) statistical analysis was performed using the GeoMx DSP Suite software v3.1.0.218. For pathway analysis, Gene set enrichment analysis (GSEA) is implemented with the fGSEA package from Bioconductor using pathway groups from the Reactome database. For the bulk RNAseq, reads were demultiplexed using bcl2fastq/2.20.0 and preprocessed using fastp/0.21.0. Gene abundance was determined with Salmon/1.5.2 and quality control output was summarized using multiqc/1.11. Counts were log-transformed and normalized using the Trimmed Means of M values (TMM) from edgeR/3.34.1 for graphical analysis. Differential expression analysis was performed using limma/3.48.3. Mitochondrial gene lists were retrieved from the HUGO Gene Nomenclature Committee (HGNC). R and Python programming languages were used for data visualization. The Benjamini–Hochberg correction was used to adjust for multiple comparisons, with a corrected value of p (q value) ≤ 0.05 considered to be statistically significant.

For all experiments, each data value corresponds to a distinct individual.

For two way ANOVA $F_{(DFn, DFd)}$: Fig. 1b, IFNβ Interaction $F_{(2, 35)} = 2858$, Age $F_{(2, 35)} = 6.157$, Genotype $F_{(1, 35)} = 3.546$; IFNγ Interaction $F_{(2, 35)} = 2828$ Age $F_{(2, 35)} = 4.064$, Genotype $F_{(1, 35)} = 3.310$; CXCL10 Interaction $F_{(2, 35)} = 7.933$ Age $F_{(2, 35)} = 8.661$, Genotype $F_{(1, 35)} = 30.29$; CXCL9 Interaction $F_{(2, 35)} = 2.312$ Age $F_{(2, 35)} = 3.041$, Genotype $F_{(1, 35)} = 2.908$; CCL2 Interaction $F_{(2, 35)} = 2.346$ Age $F_{(2, 35)} = 3.257$, Genotype $F_{(1, 35)} = 3.144$. Figure 3a,

Interaction $F_{(2, 27)} = 18.30$, Age $F_{(2, 27)} = 22.22$, Genotype $F_{(1, 27)} = 44.16$. Figure 5a, Interaction $F_{(2, 35)} = 5.019$, Age $F_{(2, 35)} = 3.225$, Genotype $F_{(1, 35)} = 5.215$. Figure 6a, Interaction $F_{(15,240)} = 3.604$, Time $F_{(15, 240)} = 43.76$, Treatment $F_{(1, 16)} = 6.178$. Figure 6c, Step Sequence Regularity Interaction $F_{(5, 93)} = 7.208$, Time $F_{(5, 93)} = 16.35$, Treatment $F_{(1, 93)} = 11.90$, Sequence Pattern AB Interaction $F_{(5, 93)} = 2.417$, Time $F_{(5, 93)} = 9.430$, Treatment $F_{(1, 93)} = 9.603$, Speed Interaction $F_{(5, 95)} = 0.9178$, Time $F_{(5, 95)} = 18.23$, Treatment $F_{(1, 95)} = 6.073$, Diagonal support Interaction $F_{(5, 93)} = 0.6279$, Time $F_{(5, 93)} = 22.15$, Treatment $F_{(1, 93)} = 17.09$, Phase dispersions RF/RH Interaction $F_{(5, 95)} = 2.441$, Time $F_{(5, 95)} = 3.297$, Treatment $F_{(1, 95)} = 9.927$, Phase dispersions RF/LH Interaction $F_{(5, 93)} = 2.364$, Time $F_{(5, 93)} = 2.728$, Treatment $F_{(1, 93)} = 9.811$, FP Swing speed Interaction $F_{(5, 95)} = 2.581$, Time $F_{(5, 95)} = 10.26$, Treatment $F_{(1, 95)} = 12.36$, FP Print Area Interaction $F_{(5, 95)} = 0.9671$, Time $F_{(5, 95)} = 8.092$, Treatment $F_{(1, 95)} = 0.8639$, FP Max Intensity Mean Interaction $F_{(5, 95)} = 1.372$, Time $F_{(5, 95)} = 11.03$, Treatment $F_{(1, 95)} = 0.01179$, FP Duty Cycle Interaction $F_{(5, 95)} = 3.124$, Time $F_{(5, 95)} = 3.207$, Treatment $F_{(1, 95)} = 5.335$. Figure 6d, Interaction $F_{(5, 96)} = 1.927$, Time $F_{(5, 96)} = 25.45$, Treatment $F_{(1, 96)} = 16.86$. Supplementary Fig. 1 F4/80 Interaction $F_{(2, 23)} = 15.26$, Age $F_{(2, 23)} = 15.41$, Genotype $F_{(1, 23)} = 16.52$, CD4 Interaction $F_{(2, 23)} = 15.62$, Age $F_{(2, 23)} = 13.54$, Genotype $F_{(1, 23)} = 25.75$, CD8 Interaction $F_{(2, 23)} = 37.06$, Age $F_{(2, 23)} = 39.85$, Genotype $F_{(1, 23)} = 54.64$, B220 Interaction $F_{(2, 23)} = 10.05$, Age $F_{(2, 23)} = 9.761$, Genotype $F_{(1, 23)} = 22.18$. Supplementary Fig. 6d Interaction $F_{(5, 120)} = 18.31$, Time $F_{(5, 120)} = 24.09$, Treatment $F_{(1, 120)} = 156.3$. Supplementary Figure 6f Interaction $F_{(3, 80)} = 10.06$, Time $F_{(3, 80)} = 12.40$, Treatment $F_{(1, 80)} = 110.4$. Supplementary Figure 11a Interaction $F_{(5, 156)} = 5.221$, Time $F_{(5, 156)} = 13.43$, Treatment $F_{(1, 156)} = 101.5$. Supplementary Figure 11c Interaction $F_{(2, 178)} = 0.7407$, Time $F_{(2, 78)} = 6.343$, Treatment $F_{(1, 78)} = 5.557$.

## Reporting summary

Further information on research design is available in the Nature Portfolio Reporting Summary linked to this article.

## Data availability

The mass spectrometry proteomics data have been deposited to the ProteomeXchange Consortium via the PRIDE[73] partner repository with the dataset identifier PXD048004. Nanostring spatial transcriptome data are available from the Gene Expression Omnibus (GEO) under accession number GSE262352. The human transcriptome data are available from the GEO under accession number GSE220915. Source data are provided with this paper.

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

## Acknowledgements

This work was supported by the Association Française contre les Myopathies (Grant CLT/SS/19458), and the European Union and Normandie Regional Council. Europe gets involved in Normandie with European Regional Development Fund (ERDF). This study was funded, in part, by the Intramural Research Program of the National Institute of Arthritis and Musculoskeletal and Skin Diseases, National Institutes of Health. We would like to thank C. Theresine and M. DiGiovanni (PRIMACEN) for their help with the oxidative stress qPCR analysis and J.R. Colin for his help with the histological analysis. We thank Drs. Jean Claude do Rego and Jean Luc do Rego from the Behavioral Analysis Platform SCAC (University of Rouen Normandy, France) for their assistance with locomotor activity studies with the Catwalk XT apparatus. We would like to thank Drs. David Scoville and Ana Ortalli (Nanostring) for their assistance with the Nanostring spatial transcriptome analysis.

## Author contributions

O.B. and C.A. conceived the project design and initiated the project. C.A. performed animal treatments, animal locomotor activity assessments, histological and molecular studies, and spatial transcriptome analysis. C.A. and R.Z. performed disease scoring. C.A. and I.R-J. measured ROS production by EPR. L.J. and R.Z. assured animal breeding, genotyping, and maintenance. C.A., G.B., L.D., and R.Z. performed mouse dissections and tissue collections. C.G. and P.C. performed the proteome experiments and analysis. I. P-F. and A.Mam. performed the human transcriptome analysis. I.P-F. and M.G. performed the in vitro myoblast/myotube studies. L.Deb., B.G., and A.M. performed the mitochondrial respiration ex vivo experiments and analysis. C.A., S.B., and D.G. performed the electron microscopy studies. O.B., C.A., I.P-F., C.G., G.B., L.D., P.C., M.G., L.Deb., L.J., S.B., D.G., R.Z., I.R-J., B.G., C.B., A.Mam., and A.M. contributed to the discussions. C.A. and O.B. wrote the paper with inputs from the other authors and I.P-F., A.Mam., A.M., C.B., C.G., and P.C. revised and edited the manuscript. O.B., C.A., I.P-F., C.G., G.B., L.D., P.C., M.G., L.Deb., L.J., S.B., D.G., R.Z., I.R-J., B.G., C.B., A.Mam., and A.M. reviewed the final version of the paper. O.B. supervised the research.

## Competing interests

O.B. received research funding and/or honorarium from argenx, BMS, CSL Behring, Egle TX, OGD2, and UCB. The remaining authors declare no competing interests.
