## [Peer Review File · Nature Communications]

IFN γ causes mitochondrial dysfunction and oxidative stress in myositisREVIEWER COMMENTS

Reviewer #1 (expert in neuromuscular disorders):

The authors sought to explore the mechanisms that contribute to myositis pathogenesis in an Icos^{-/-} NOD mouse model of myositis. The authors report that Icos^{-/-} NOD mice develop a myositis-like phenotype and show mitochondrial abnormalities and metabolic dysfunction that was ameliorated by treatment with NAC. They found support in human myositis transcriptome analyses. The authors state that their data demonstrates a connection between inflammation, mitochondrial dysfunction, and ROS contributing to myositis disease. The paper was well written, the methods were sound, and the authors used a variety of assays including functional tests, histology, proteomic and transcriptomic analysis, gene expression, among others to thoroughly investigate their research questions. This adds to the field in that it provides further support for mitochondrial dysfunction/ROS as potential therapeutic targets for myositis. I have only minor comments:

1. Figure 6: Which of these correlations are significant? It is not listed in the figure caption and the text in the results section states that there are some negative and positive correlations. Given that a major claim of the authors is the correlation between IFN γ signature genes and mitochondrial dysfunction in myositis, it should be made clearer.
2. The authors should make it clear that the IFN γ and NAC treatments (asymptomatic and symptomatic) were all started at different ages. Why was IFN γ treatment initiated at week 21? How does this impact comparisons?
3. All bar graphs should be replaced with graphs showing individual data points.
4. The authors did not discuss sex as a biological variable. It is understandable why the authors used only females; however, this should be briefly explained. Have they compared sex differences in previous studies using this model?
5. The authors discuss no limitations.
6. Figures: In several figures, the titles containing symbols (e.g., gamma and beta) are obstructed.

Reviewer #2 (expert in neuromuscular disorders and proteomics):

In their article entitled "A pathogenic role for mitochondrial dysfunction and oxidative stress in myositis", Abad and colleagues introduce a mouse model (Icos^{-/-} NOD mice) for myositis and performed comprehensive studies to elucidate the myopathology along with the underlying pathophysiology. To do so, proteomic studies were carried out in addition to spatial transcriptomics, general transcript studies and microscopic studies including electron microscopy. The applied techniques are all sound and provide significant insights into the underlying pathophysiology by hinting toward a profound vulnerability of muscle cell mitochondria based on oxidative stress burden. The impact of IFN γ in this context is further investigated by making use of a human in vitro model and relevance of data obtained from the study on the mouse model are backed up with transcript data obtained on muscle biopsy specimen derived from DM patients with confirmed serotypes.

The study is well designed and provides new significant insights into the field of myositis and along this line introduces a suitable animal model. Thus, I am convinced that the data of great interest for a broad readership and should be published in Nature Communications after some points of concern have been addressed:

Major points:

- Authors should provide more information regarding the transcript dysregulations across the different serotypes; is there a major affection of one certain serotype? Were paediatric cases suffering from a juvenile form of DM included? These are important information regarding future avenues in patient stratification
- Electron microscopy focussing on mitochondrial ultra-structures in IFN γ treated human myoblasts should be included
- Authors should mention on potential proteomic or transcript findings hinting toward affection of

peroxisomes

- Given that genetic data on sIBM unveiled FYCO1 as a potential modifier (doi: 10.1002/ana.24847), I wonder, if genetic data focussing on potential variants/ SNPs in ICOS could also be provided for DM cases included in this study. Variants affecting the coding regions should even be visible in the transcript data
- Discussion section should be shortened

Minor points:

- Page 3: correct term is: Endoplasmic Reticulum stress
- Transcripts should be named in italic letters throughout the manuscript

Reviewer #3 (expert in nerve/muscle axis and muscle anatomy):

In this manuscript, Abad and colleagues explore the interconnection between mitochondrial dysfunction, ROS, and inflammation in idiopathic inflammatory myositis (IIM). Using the previously presented Icos^{-/-}-NOD mouse model for IIM (Bourdenet et al 2022), the authors use proteomic and spatial transcriptomic approaches to link inflammation to the metabolic alteration in muscle fibers. Indeed, treatment aimed at reducing ROS improves pathology, reduces inflammation, and preserves mitochondrial integrity. The authors also identify INF γ as a possible mediator linking inflammation to mitochondrial dysfunction.

Given the limited knowledge of this disease and the lack of selective treatment, the manuscript could represent a step toward clinical treatment of the disease. However, some weaknesses need to be resolved before publication.

The authors used the NOD mouse (background of the IIM model) as a control. However, it should not be forgotten that this animal develops another disease, diabetes, which could alter some parameters. Therefore, I think it is important to include a control consisting of non-diabetic animals as a reference of healthy individuals in the analyses. In line with this observation, it is important to confirm the main observations in another canonical model of myositis.

In addition, there are still several issues that the authors need to address:

- 1) It is unclear to this reviewer whether or not inflammation precedes metabolic changes in Icos^{-/-}-NOD muscle. As partially performed in the previous article, the author should quantify and identify different immune cell populations during disease progression and as a consequence of NAC (and/or anti-IFN γ) treatment. In addition, it might be of interest to reanalyze the spatial transcriptomic data of inflammatory areas to specifically identify the immune cell population near the damaged fibers.
- 2) Metabolic changes evidenced by COX, NADH-TR and SDH staining need to be quantified. In addition, whether the metabolic alteration of muscle fibers is associated with a switch in myosin type, as observed in other muscle diseases, should be investigated. Finally, it might be interesting to understand (also using ST analysis) whether glycolytic versus oxidative fibers are more susceptible to muscle damage in this model.
- 3) In previous work, muscle fibrosis appears in Icos^{-/-}-NOD muscle due to myositis. In this regard, it should be analyzed whether NAC (and/or anti-IFN γ) treatment prevents fibrosis deposition?
- 4) In the previous article, the authors identify autoantibody against TNN3 as the cause of IIM in Icos^{-/-}-NOD mice. Does NAC (and/or anti-IFN γ) treatment then prevent Ab-TNN3 production or merely mitigate muscle damage?

Minor:

- 1) Oxidative stress-associated apoptosis was evidenced in Icos^{-/-}-NOD muscle. Using immunofluorescence, the authors should identify the population of cells (fibers or other resident muscle cells) that go into apoptosis.
- 2) Does preventing the progression of myositis through inhibition of ROS (and/or anti-IFN γ) processing promote or not the recurrence of diabetes?
- 3) Four different housekeeping genes for RT-PCR are presented in the methods. It is not clear in which analysis one is used rather than another and why.

RESPONSE TO REVIEWERS' COMMENTS

Reviewer #1 (expert in neuromuscular disorders):

The authors sought to explore the mechanisms that contribute to myositis pathogenesis in an Icos $-/-$ NOD mouse model of myositis. The authors report that Icos $-/-$ NOD mice develop a myositis-like phenotype and show mitochondrial abnormalities and metabolic dysfunction that was ameliorated by treatment with NAC. They found support in human myositis transcriptome analyses. The authors state that their data demonstrates a connection between inflammation, mitochondrial dysfunction, and ROS contributing to myositis disease. The paper was well written, the methods were sound, and the authors used a variety of assays including functional tests, histology, proteomic and transcriptomic analysis, gene expression, among others to thoroughly investigate their research questions. This adds to the field in that it provides further support for mitochondrial dysfunction/ROS as potential therapeutic targets for myositis. I have only minor comments:

We are delighted that this reviewer found that our work “adds to the field” and provides “further support [...] to the identification of...” “therapeutic targets for myositis”. We respond below to the reviewer’s minor comments.

1. Figure 6: Which of these correlations are significant? It is not listed in the figure caption and the text in the results section states that there are some negative and positive correlations. Given that a major claim of the authors is the correlation between IFN γ signature genes and mitochondrial dysfunction in myositis, it should be made clearer.

We now provide the complete list of p-values in the new Supplementary Tables 2 and 3 (cited on page 14).

2. The authors should make it clear that the IFN γ and NAC treatments (asymptomatic and symptomatic) were all started at different ages. Why was IFN γ treatment initiated at week 21? How does this impact comparisons?

We chose to start anti-IFN γ treatment at week 21 in order to be preventive, since T-cell infiltrates were clearly detectable around week 25. Since proteome analysis started to show minimal metabolic changes in the muscle as early as 8 weeks of age, way before overt ROS production, we wanted to make sure that the preventive treatment was initiated early enough. We believe that this does not impact our conclusions, since we do not directly compare these two therapeutic regimens targeting very different processes. We modified the text to make this point clearer (pages 11 and 12).

3. All bar graphs should be replaced with graphs showing individual data points.

This has been done

4. The authors did not discuss sex as a biological variable. It is understandable why the authors used only females; however, this should be briefly explained. Have they compared sex differences in previous studies using this model?

We chose to use only females because the incidence of spontaneous myopathy (70%) is significantly higher than in males (20%). This is now briefly explained in the material and methods section (page 21) as well as in the Reporting Summary. We have compared sex differences in our previous study using this model (Prevot, N. *et al* Eur J Immunol. 2010 Aug;40(8):2267-76. doi: 10.1002/eji.201040416. PMID: 20544729).

5. The authors discuss no limitations.

Limitations have now been discussed in the manuscript (page 20).

6. Figures: In several figures, the titles containing symbols (e.g., gamma and beta) are obstructed.

We respectfully point out that in our pdf versions the symbols are legible. We will check that this not appear in further steps.

Reviewer #2 (expert in neuromuscular disorders and proteomics):

In their article entitled “A pathogenic role for mitochondrial dysfunction and oxidative stress in myositis”, Abad and colleagues introduce a mouse model (Icos^{-/-} NOD mice) for myositis and performed comprehensive studies to elucidate the myopathology along with the underlying pathophysiology. To do so, proteomic studies were carried out in addition to spatial transcriptomics, general transcript studies and microscopic studies including electron microscopy. The applied techniques are all sound and provide significant insights into the underlying pathophysiology by hinting toward a profound vulnerability of muscle cell mitochondria based on oxidative stress burden. The impact of IFN γ in this context is further investigated by making use of a human in vitro model and relevance of data obtained from the study on the mouse model are backed up with transcript data obtained on muscle biopsy specimen derived from DM patients with confirmed serotypes.

The study is well designed and provides new significant insights into the field of myositis and along this line introduces a suitable animal model. Thus, I am convinced that the data of great interest for a broad readership and should be published in Nature Communications after some points of concern have been addressed:

We are delighted that this reviewer found this study “well designed” and “provides new significant insights into the field of myositis”.

Major points:

- Authors should provide more information regarding the transcript dysregulations across the different serotypes; is there a major affection of one certain serotype? Were paediatric cases suffering from a juvenile form of DM included? These are important information regarding future avenues in patient stratification

Interestingly, we found no relevant changes of mitochondrial gene expression among serologic groups, indicating that the metabolic defects observed is a general pathophysiological mechanism of the disease rather than ascribed to a specific myositis subset. This has been now reported in the manuscript (page 15) and a new graph illustrating this point has been added as a supplementary Figure (Supplementary Figure 16). Regarding IFN expression according to IIM serotype, we have previously reported these data in Pinal-Fernandez I, et al. *Neurology*. 2019;93:e1193-e1204. (doi: 10.1212/WNL.0000000000008128). No juvenile dermatomyositis cases were included in this study, which is now indicated in the Material and Methods section (page 30).

- Electron microscopy focusing on mitochondrial ultra-structures in IFN γ treated human myoblasts should be included

We thank the reviewer for this comment. We have treated human myoblasts (LHCN-M2 cell line) with IFN γ and performed electron microscopy. We found that exposure to this cytokine induced mitochondrial abnormalities (enlargement, mitophagy), which were prevented by treatment with a JAK inhibitor (ruxolitinib or baricitinib). This supports a direct deleterious effect of this cytokine on muscle mitochondria. These new data have been now integrated in the text (page 15) and in Figure 6.

- Authors should mention on potential proteomic or transcript findings hinting toward affection of peroxisomes

We agree with the reviewer that this is an important question since multiple studies show a close relationship between peroxisomes with mitochondria. Thus, we have further analyzed our data looking closer at peroxisome proteins and genes. Of the 52 proteins identified under the term “peroxisome” by IPA, we found several proteins downregulated in the muscle of *Icos*^{-/-} NOD mice at 8, 25 and 35 week of age. These results are now presented in Supplementary Figure 18.

Concerning the Nanostring data, the expression of a few genes related to the peroxisome was altered in myofibers from *Icos*^{-/-} NOD mice, being mostly downregulated (see Supplementary Figure 18). The most significant downregulations corresponded to *Ubb*, *Crat*, *Phyh*, *Fis1*, *Ech1* and *Ube2d3*. Many of these gene products are not exclusively related to peroxisomes and are also related to mitochondria. Nevertheless, two of them, *Phyh* (coding for phytanoyl-CoA 2-hydroxylase) and *Ech1* (coding for delta(3,5)-Delta(2,4)-dienoyl-CoA isomerase) seem to be specific of this organelle in humans. Interestingly, we found that the expression of these two genes was downregulated in human DM biopsies (see Supplementary Figure 18). This has now been mentioned in the discussion (page 19).

- Given that genetic data on sIBM unveiled FYCO1 as a potential modifier (doi: 10.1002/ana.24847), I wonder, if genetic data focussing on potential variants/ SNPs in ICOS could also be provided for DM cases included in this study. Variants affecting the coding regions should even be visible in the transcript data

Our RNA-seq is not the ideal technique to answer this question. Two large IIM genetic studies that include dermatomyositis patients have been performed. The first one studied a large cohort of myositis patients including dermatomyositis (n=879), juvenile dermatomyositis (n=481), polymyositis (n=931) and inclusion body myositis (n=252) patients (Rothwell et al, Ann Rheum Dis 206, 75: 1558-66). The second one included 705 adult and 473 juvenile dermatomyositis patients (Arthritis Rheum. 2013; 65:3239-47). None of these two studies revealed an association of *Icos* gene variants/SNPs with IIMs. In addition, we performed a search within large public gene expression databases (DisGeNET, Disease and NHGRI-EBI GWAS Catalog), and we found no reported *Icos* variants/SNPs associated to dermatomyositis (or other myositis type).

References

DisGeNET (Piñero J, Ramírez-Anguita JM, Saüch-Pitarch J, Ronzano F, Centeno E, Sanz F, et al. The DisGeNET knowledge platform for disease genomics: 2019 update. Nucleic Acids Res. 2019;5:gkz1021)

Diseases (Grissa D, Junge A, Oprea TI, Jensen LJ. Diseases 2.0: a weekly updated database of disease-gene associations from text mining and data integration. Database (Oxford). 2022 Mar 28;2022:baac019. doi: 10.1093/database/baac019)

The NHGRI-EBI GWAS Catalog: knowledgebase and deposition resource (Nucleic Acids Res. 2022 Nov 9:gkac1010. doi: 10.1093/nar/gkac1010)

- Discussion section should be shortened

This has been done

Minor points:

- Page 3: correct term is: Endoplasmic Reticulum stress

This has been corrected

- Transcripts should be named in italic letters throughout the manuscript

This has been done

Reviewer #3 (expert in nerve/muscle axis and muscle anatomy):

In this manuscript, Abad and colleagues explore the interconnection between mitochondrial dysfunction, ROS, and inflammation in idiopathic inflammatory myositis (IIM). Using the previously presented *Icos*^{-/-} NOD mouse model for IIM (Bourdenet et al 2022), the authors use proteomic and spatial transcriptomic approaches to link inflammation to the metabolic alteration in muscle fibers. Indeed, treatment aimed at reducing ROS improves pathology, reduces inflammation, and preserves mitochondrial integrity. The authors also identify INF γ as a possible mediator linking inflammation to mitochondrial dysfunction. Given the limited knowledge of this disease and the lack of selective treatment, the manuscript could represent a step toward clinical treatment of the disease. However, some weaknesses need to be resolved before publication.

The authors used the NOD mouse (background of the IIM model) as a control. However, it should not be forgotten that this animal develops another disease, diabetes, which could alter some parameters. Therefore, I think it is important to include a control consisting of non-diabetic animals as a reference of healthy individuals in the analyses.

In our first experiment, we used *Icos*^{+/+} NOD but also non-diabetes prone mice (Balb/c) mice. Importantly, we found no differences in muscle strength between *Icos*^{+/+} NOD and Balb/c, nor in muscle ROS production (Figure below).

Figure 1. Muscle strength after sciatic nerve stimulation (strength *in situ*) and muscle ROS production measured by electron paramagnetic resonance (EPR)

Furthermore, in the experiments reported in this paper, we had monitored the presence of glucose in the urine by using test sticks, and we only used as controls *Icos*^{+/+} NOD mice that did not develop diabetes. Given these facts, we consider that *Icos*^{+/+} NOD are the most suitable control since they share the same genetic background. We apologize because this was not indicated in the original manuscript, but this point is now clearly indicated (material and methods section, page 21).

In line with this observation, it is important to confirm the main observations in another canonical model of myositis.

In a previous study, we have already reported increased oxidative stress and the presence of mitochondrial damage in the canonical inflammatory EAM model (see Figure from this paper below).

Figure 2. Extracted from Meyer A, et al Acta Neuropathol. 2017 Oct;134(4):655-666. doi: 10.1007/s00401-

017-1731-9. Mitochondria respiration rates in the presence of glutamate, malate, and saturating amounts of ADP (V_{max}) (e), TMPD + ascorbate (V_{TMPD}) (f), and respiratory control ratio (V_{max}/V_0 , V_0 representing respiration rates in the presence of glutamate and malate without ADP) (g) in gastrocnemius and soleus muscle of control (white bars) and EAM mice treated (dashed bars) or not (black bar) with NAC (n = 12 per group). h Reactive oxygen species levels in gastrocnemius and quadriceps muscles. * $p < 0.05$ vs. controls, # $p < 0.05$ vs. NAC-treated EAM, by one-way ANOVA and post hoc Newman–Keuls range test

These findings in another model of myositis are fully consistent with the more detailed results presented herein in *Icos*^{-/-} NOD mice. We have now indicated more clearly in the revised manuscript that our findings are corroborated by another model of myositis (page 20).

In addition, there are still several issues that the authors need to address:

1) It is unclear to this reviewer whether or not inflammation precedes metabolic changes in *Icos*^{-/-}-NOD muscle. As partially performed in the previous article, the author should quantify and identify different immune cell populations during disease progression and as a consequence of NAC (and/or anti-IFN γ) treatment. In addition, it might be of interest to reanalyze the spatial transcriptomic data of inflammatory areas to specifically identify the immune cell population near the damaged fibers.

We performed immunofluorescence staining of immune cells in the muscle sections from all our protocols (kinetics, preventive and curative NAC and anti-IFN γ), as requested. We did not find immune cell infiltrates at 8 weeks of age (asymptomatic mice), whereas some minimal changes at proteome level were already observed. Since we previously demonstrated the T-cell dependency of disease by adoptive transfer experiments (Briet *et al*, Front Immunol. 2017;8:287) and that anti-IFN γ totally abolished disease, this suggests that systemic rather than local activation of the immune system precedes the metabolic changes. Regarding the treatment studies, we found that NAC treatment reduced macrophage, CD4, CD8 and B cell infiltration, with a more significant decrease observed in the preventive setting. The anti-IFN γ treatment also reduced infiltration by all these cell populations. These new data have now been added in Supplementary Figures 1, 9, 11 and 13.

Whereas IF showed the predominance of macrophages and CD4 T cells in inflammatory infiltrates, deconvolution of the spatial transcriptome data in our study using the SpatialDecon script from Nanostring DSP and a publicly available large immune cell profile matrix (ImmuneAtlas-ImmGen.csv from Yoshida *et al*, Cell. 2019;176(4):897-912.e20 available at <https://github.com/Nanostring-Biostats/CellProfileLibrary/tree/master/Mouse>) allowed to further identify the presence of other immune cell populations as seen in Figure 3 below. Among others, we note the presence of natural killer (NK), neutrophils and innate lymphoid cells (ILC), notably ILC3, which are related to inflammation. Further study of these populations is out of the scope of this manuscript but may be the object of future studies.

Figure 3. Deconvolution results of immune cell transcriptome data. A table with a description of the most upregulated genes is shown on the right.

2) Metabolic changes evidenced by COX, NADH-TR and SDH staining need to be quantified.

We observed a reduction of the activity for all these three enzymes in diseased mice, that was restored by treatment. Indeed, a loss of fibers with the highest COX activity was evident. This has been now quantified and these data has been added in Figure 2, Figure 5, Supplementary Figure 5 and Supplementary Figure 7. The patterns of COX, NADH-TR and SDH staining were identical among each other, and thus we do not expect our findings with COX quantification to differ for the other enzymes.

In addition, whether the metabolic alteration of muscle fibers is associated with a switch in myosin type, as observed in other muscle diseases, should be investigated. Finally, it might be interesting to understand (also using ST analysis) whether glycolytic versus oxidative fibers are more susceptible to muscle damage in this model.

Type 1 fibers (oxidative fibers) express MYH7 and MYL3 whereas type 2 fibers (glycolytic fibers) express MYH1, MYH2 and MYH4. At spatial transcriptomic level, we observed a general reduction in the levels of all myosins detected (*Myh1*, *Myh2*, *Myh4*), particularly when in close contact with the immune cell infiltrates (*Myh7* was not detectable in the AOI analyzed), with no skewing towards one or another fiber type. Consistently, there is a significant and similar reduction of both oxidative and glycolytic pathways (see Table below).

	PROX vs Control	ADJ vs Control	PROX vs ADJ
Glycolysis			
NES	3.2267	3.3459	2.454
Adjusted p value	0.0001	0.0056	0.0002
Respiratory chain			
NES	4.7624	3.7842	2.4517
Adjusted p value	0.0001	0.0003	0.0003

This was corroborated by immunofluorescence data (n=3 *Icos*^{+/+} NOD mice and n=5 *Icos*^{-/-} NOD mice) : there was no significant differences in the proportions of MYH2+ (13.27 ± 2.99% in *Icos*^{+/+} NOD vs. 14.22 ± 2.91% *Icos*^{-/-} NOD mice), MYH4+ (77.65 ± 3.90% in *Icos*^{+/+} NOD vs. 86.43 ± 1.67% *Icos*^{-/-} NOD mice) and MYH7+ (2.88 ± 1.17% in *Icos*^{+/+} NOD vs. 2.69 ± 0.78% *Icos*^{-/-} NOD mice) fibers.

Thus, our data does not suggest a major switch in glycolytic versus oxidative fibers.

3) In previous work, muscle fibrosis appears in *Icos*^{-/-}-NOD muscle due to myositis. In this regard, it should be analyzed whether NAC (and/or anti-IFN γ) treatment prevents fibrosis deposition?

We thank the reviewer for raising this point. We performed Sirius red staining of muscle sections from NAC (preventive and curative) and anti-IFN γ protocols in our study, and found that all these treatments diminished fibrosis in a significant manner. A new supplementary Figure has been added gathering these results (Supplementary Figure 10) and is cited on pages 12 and 13.

4) In the previous article, the authors identify autoantibody against TNN3 as the cause of IIM in *Icos*^{-/-}-NOD mice. Does NAC (and/or anti-IFN γ) treatment then prevent Ab-TNN3 production or merely mitigate muscle damage?

We have measured the levels of anti-TNNT3 antibodies in the sera from all our protocols, as requested. Interestingly, we found a significant decrease with the preventive NAC treatment but not with the curative treatment. This result has been now included in Supplementary Table 1 and is cited on page 13.

Regarding the anti-IFN γ protocols, a fewer number of serum samples was available. A trend to a reduction was observed but was not significant (not shown).

Minor:

1) Oxidative stress-associated apoptosis was evidenced in *Icos*^{-/-}-NOD muscle. Using immunofluorescence, the authors should identify the population of cells (fibers or other resident muscle cells) that go into apoptosis.

Indeed, our oxidative stress bulk transcriptome analysis had shown an increase in Caspase 3 and Bax expressions, which suggests the presence of apoptosis. Nevertheless, we have now performed immunofluorescence studies by using TUNEL, cleaved Caspase 3 and Bax antibodies, and the results are inconclusive (not shown). It is possible that upregulation of these proteins at transcriptional level does not ultimately lead to apoptosis.

2) Does preventing the progression of myositis through inhibition of ROS (and/or anti-IFN γ) processing promote or not the recurrence of diabetes?

Again, this is a very interesting question. We have now measured the level of glucose in the sera of all the mice in our protocols, and found that none of them treated (or untreated) exhibited hyperglycemia (compared to a positive control consisting of mice treated with alloxan, a well-known model of diabetes). Therefore, it doesn't seem that these treatments lead to overt diabetes. These new data have been added as Supplementary Table 4 and commented in the discussion (page 16).

3) Four different housekeeping genes for RT-PCR are presented in the methods. It is not clear in which analysis one is used rather than another and why.

We apologize that this was not clear. The transcriptome analysis (multiplex qPCR) of oxidative stress genes was performed by our institutional facility using a premade panel that included four different housekeeping genes (*Rplp0*, *Tfrc*, *Hsp90ab1*, and *Nono*). For the other qPCR measurements that were performed in our lab, we selected only one housekeeping gene (*Hsp90ab1*) since, in a preliminary experiment, we found no difference in gene expression when normalized vs. this gene or the average of *Rplp0*, *Tfrc*, *Hsp90ab1*, and *Nono*. This has been specified in the material and methods (page 26).

REVIEWERS' COMMENTS

Reviewer #1 (Remarks to the Author):

The authors have addressed all reviewer comments to my satisfaction.

Reviewer #2 (Remarks to the Author):

The authors sufficiently addressed all my points of concern!
Congratulations to this very nice manuscript!

Reviewer #3 (Remarks to the Author):

The new version of the manuscript appears to be improved.
The authors accurately answered all the criticisms raised by the reviewers. The authors supplemented the article with new evidence and discussion to support the conclusions of the study. Therefore, I believe that the manuscript is ready for publication in this journal.